

# Exclusive use and evaluation of inheritance metrics viability in software fault prediction—an experimental study

Syed Rashid Aziz[1], Tamim Ahmed Khan[1] and Aamer Nadeem[2]

[1] Department of Software Engineering, Bahria University, Islamabad, Pakistan
[2] Department of Software Engineering, Capital University of Science and Technology, Islamabad, Pakistan

## ABSTRACT

Software Fault Prediction (SFP) assists in the identification of faulty classes, and software metrics provide us with a mechanism for this purpose. Besides others, metrics addressing inheritance in Object-Oriented (OO) are important as these measure depth, hierarchy, width, and overriding complexity of the software. In this paper, we evaluated the exclusive use, and viability of inheritance metrics in SFP through experiments. We perform a survey of inheritance metrics whose data sets are publicly available, and collected about 40 data sets having inheritance metrics. We cleaned, and filtered them, and captured nine inheritance metrics. After preprocessing, we divided selected data sets into all possible combinations of inheritance metrics, and then we merged similar metrics. We then formed 67 data sets containing only inheritance metrics that have nominal binary class labels. We performed a model building, and validation for Support Vector Machine(SVM). Results of Cross-Entropy, Accuracy, F-Measure, and AUC advocate viability of inheritance metrics in software fault prediction. Furthermore, ic, noc, and dit metrics are helpful in reduction of error entropy rate over the rest of the 67 feature sets.

# INTRODUCTION

Object-Oriented software development is a widely used software development technique. It is not possible to produce a completely fault-free software system. The failure rates in a thousand lines of code for industrial projects are 15–50, 10–20 for Microsoft (*McConnell, 2004*) and specifically in Windows 2000, which comprises thirty-five million Lines of Code(LOC) recorded sixty-three thousand errors (*Foley, 2007*). Residual errors may cause failure since many types of errors have been newly detected (*Bilton, 2016*; *Osborn, 2016*; *Grice, 2015*).

Early detection of faults may save time, costs, and decrease the software complexity since it is proportional to the testing. Extensive testing is required to locate all remaining errors. The extensive tests are impossible (*Jayanthi & Florence, 2018*; *Kaner, Bach & Pettichord, 2008*). This is why the cost of testing at times elevated to over 50% of the total software development cost (*Majumdar, 2010*). This number may rise to seventy-five

Corresponding author
Syed Rashid Aziz,
rashid_cbr@yahoo.com

percent as reported by IBM (*Hailpern & Santhanam, 2002*). Software testing is essential and inevitable to produce software without errors. Thorough testing of entire classes with limited manpower is a challenging task. It is more feasible to identify faulty classes and test them to produce software with good quality. It is observed those faults are not uniformly dispersed throughout the software product. Certain classes are more faulty as compare to others and are clustered in a limited number of classes (*Sherer, 1995*). Research shows that errors are limited to 42.04% of the entire components in a software project (*Gyimothy, Ferenc & Siket, 2005*). Similarly, the findings of *Ostrand, Weyuker & Bell (2004)* revealed that only 20% of the entire software components are faulty in a project.

In software engineering, presently numerous prediction approaches exist that include reuse prediction, test effort prediction, security prediction, cost prediction, fault prediction, quality prediction, and stress prediction (*Catal, 2011*). SFP is an up-and-coming research area, in which defective classes are discovered in the initial period of project development (*Menzies, Greenwald & Frank, 2007*; *Jing et al., 2014*; *Seliya, Khoshgoftaar & Van Hulse, 2010*) with machine learning (*Catal, 2011*; *Malhotra, 2015*). It is assumed that with the help of metrics, fault prediction model may be constructed to calculate coupling, inheritance, cohesion, size, and complexity of software (*Chen & Huang, 2009*; *Chidamber & Kemerer, 1994*; *Li & Henry, 1993*; *Basili, Briand & Melo, 1996*; *Malhotra & Jain, 2012*).

The fault prediction process typically includes the training and prediction phases. In the first phase of training, a prediction model is constructed utilizing software metrics at class or method level together with fault information associated with each module of the software. Later, the said model is employed on a newly developed software version for faults prediction.

We utilize methods of classification for labeling classes into fault-free or faulty classes by using software metrics and fault information. Quality of software is improved by detecting classes with faults within the software using fault prediction models. The model performance has an effect on the modeling technique (*Elish & Elish, 2008a*; *Catal & Diri, 2009*) and metrics (*Briand et al., 2000*; *Ostrand, Weyuker & Bell, 2005*). Several researchers have developed and validated fault prediction models based on statistical techniques or machine learning. They have used software metrics, data sets, and feature reduction methods to achieve improvement in the performance of model.

The software development industry is utilizing Object-Oriented paradigm; however, the usage of metrics at class-level in fault prediction is less common while comparing with other types of metrics (*Catal & Diri, 2009*). Our literature review findings show that method level software metrics employing machine learning techniques are in use by the researches (*Catal, 2011*). It is beneficial to use publicly available data sets for SFP since they help to develop verifiable and repeatable models (*Catal & Diri, 2009*). It is pertinent to mention here that models of machine learning have superior precision and accessibility as comparing it with expert opinion-based approaches or statistical methods.

Besides others, inheritance is a prominent feature of Object-Oriented programming. Inheritance is the class's ability to obtain the properties of some other class. It is divided into several types, namely single, multiple, multi-level, and so on. Object-Oriented supports reuse in three different ways (*Karunanithi & Bieman, 1993*), where the primary technique

of reuse is through inheritance. It is the relationship between classes, in which a class object gets properties from other classes (*Rosenberg, 1998*). Forming classes into a classification hierarchy, it provides an additional dimension to the encapsulation of the abstract data types. It allows classes to inherit methods and attributes from other classes (*Breesam, 2007*). Use of inheritance reduces the essential software maintenance cost and testing burden (*Chidamber & Kemerer, 1991*; *Booch, 1991*; *Dale & Van Der Zee, 1992*; *Fenton, 1994*). Software reuse through inheritance yields more maintainable, comprehensible, and reliable software (*Chidamber & Kemerer, 1991*).

We validated inheritance metrics impact in SFP by experiments, in which we used Artificial Neural Network (ANN) for the construction of models. We used Accuracy, Recall, True negative rate (TNR), Precision, and F1-Measures as performance measures (*Aziz, Khan & Nadeem, 2019*). The outcomes indicated the satisfactory influence of metrics of inheritance on software fault prediction. The professionals associated with testing may securely utilize metrics of inheritance for fault prediction in software projects. A higher value of inheritance is also undesirable, as it might introduce faults in the software.

Generally utilized metrics of inheritance include Number of Children (noc), Depth of Inheritance (dit), Total Progeny Count (tpc), Class-to-leaf depth (cld), Total Ascendancy Count (tac), Number of Descendant Classes (ndc), Number of Ancestor Classes (nac), Number of Overridden Methods (norm), Number of Methods Inherited (nmi) and Number of Attributes Inherited (nai), Average Inheritance Depth (aid), Number of children in the hierarchy (nocc), Maximum DIT (MaxDit), Specialization Ration (SRatio), Total length of Inheritance chain(tli), Reuse Ration (RRatio), Method Inheritance Factor (mif), Attribute Inheritance Factor (aif), Number of Attributes Inherited in Hierarchy (naih), Specialization Index (si) and Number of Methods Inherited in Hierarchy (nmih) (*Reddy & Ojha, 2018*).

It was noticed during the survey on SFP that researchers have already performed so much work on the metrics of Object-Oriented software for example cohesion, coupling, etc. All these are utilized either individually or in a group with other metrics. The C&K metrics set is an example where these are used in a group and commonly recognized in the research arena. However, it was perceived that inheritance metrics are utilized collectively in C&K suite, but solo use and assessment of inheritance metrics is neglected. This prompts the performance of experimentation to concentrate explicitly on inheritance aspect to demonstrate the viability of its metrics in the perspective of SFP.

This paper is a novel representation of the following:
1. Exploring inheritance metrics having publicly available data sets.
2. Exclusive experimental evaluation of the inheritance metrics on as large as 40 data sets.
3. An empirical validation of exclusive viability of inheritance aspect on software fault prediction.
4. Across the data sets/products software fault prediction.

The paper is categorized into sections where 'Theoretical Background' explains theoretical context of SFP, Inheritance, Inheritance metrics, and their usage and data sets in SFP. The literature review explaining inheritance in SFP is presented in 'Literature Review'. 'Methodology' explains experimental approach utilized in this study and an experimental

assessment of the inheritance measures. Finally, 'Experiment and Results' explains the threats to validity, conclusion remarks and future directions of the investigation.

## THEORETICAL BACKGROUND

### Software fault prediction

The process of SFP typically includes two phases training and prediction. In the training phase method or class level software metrics with faulty data associated with the individual module of the software are used to construct a prediction model. Later, this model is used for the prediction of fault-prone classes in a newer version of the software. Fault prediction is beneficial to improve software quality and reduce testing costs. Moreover, it enables testing teams to limit testing on fault-prone classes only. In software, faults prediction theoretically established the target to grasp which parts may require concentration. Several SFP methods have been used (*Rathore & Kumar, 2017a*; *Catal, 2011*), which share three main ingredients (*Beecham et al., 2010*); Feature set, Class label, and Model.

The feature set consists of the metrics from a software artifact. It is believed that these are decent class label predictors. The metrics are classified into product, project, and process metrics. Out of these product metrics are mostly utilized (*Gómez et al., 2006*). The product metrics are further grouped into class, method, and file levels. Generally, 60% metrics at method-level are applied after that 24% metrics at class-level (*Catal & Diri, 2009*). Product metrics also comprise design, code, volume, and complexity metrics. The fault prediction model performance is so much dependent on these metrics. Scholars have evaluated the usage frequency of metrics in (*Malhotra, 2015*; *Catal & Diri, 2009*; *Beecham et al., 2010*; *Radjenović et al., 2013*; *Gondra, 2008*; *Chappelly et al., 2017*; *Nair et al., 2018*), where the utmost commonly applied software product metrics in software fault prediction are Halstead (*Halstead, 1977*), McCabe (*McCabe, 1976*), LoC in structural programming, and C&K metrics suite (*Chidamber & Kemerer, 1991*) in Object-Oriented paradigm. These metrics have became the de-facto standard metrics in SFP. PROMISE (*Boetticher, 2007*) and D'Ambros (*D'Ambros, Lanza & Robbes, 2010*) are more often used data sets repositories containing these metrics. These repositories contain data sets of about 52% of the studies published after 2005 (*Malhotra, 2015*). Since these data sets are available publicly therefore they are used very frequently. The second reason is the absence of fault data of industrial projects of software.

The second very significant element is the class label in software fault prediction, which contains metrics actual value. Within the software fault prediction domain, fault free or faulty are shown as continuous or nominal-binary or to point total faults in an occurrence. Though, usage of continuous labels present in the literature (*Rathore & Kumar, 2017b*), but leading class labels are nominal class labels in SFP (*Catal, 2011*; *Rathore & Kumar, 2017a*).

In software fault prediction, model building is a third key factor, which is a relationship between feature set and label of class. It can be applied by the use of Machine learning (ML) algorithms, statistical methods, or even expert opinion (*Catal & Diri, 2009*), where ML is an extensively utilized method for model building (*Beecham et al., 2010*). It expressively expands the accuracy of classification (*Han, Pei & Kamber, 2011*). In the SFP domain,

several ML algorithms are utilized. The performance of these ML algorithms has been compared by Malhotra, whose findings conclude that Bayesian networks and Random forest are outperformers while compared with other algorithms of ML (*Malhotra, 2015*).

The findings of literature review disclosed that 22% statistical methods are utilized (*Grice, 2015*), and 59% is machine learning. Several performance measures and machine learning methods are explored that make use of object-oriented metrics to predict faults. These studies are cataloged into multiple tables. The studies from 1990-2003 are shown in Table 1, studies from 2004 to 2007 are listed in Table 2 and finally studies from 2008 to 2020 are listed in Table 3 (*Aziz, Khan & Nadeem, 2020*).

## Software inheritance

Inheritance makes it possible to make use of the components of previous objects by recently created objects in an object-oriented paradigm. The superclass or base class is a source of inheritance and a subclass or derived class inherits from a superclass. The term main class and secondary class can also be used interchangeably for super and subclass. The sub-class can possess its components and methods in addition to inherit visible methods and properties from the main class. Inheritance offers (*Aziz, Khan & Nadeem, 2019*):

1. **Reusability:** reuse is a resource available by the inheritance where superclass's public methods are utilized into subclass without code rewriting.
2. **Overriding:** define a new behavior for a method that already exists. This happens when the class in question extends any other class and creates a method with the same signature as the "parent" class in the subclass.
3. **Extensibility:** extend the logic of the supper class according to the business logic of the sub-class.
4. **Maintainability:** it is straightforward to walk-through the source code as soon as the software program is split into portions.
5. **Data hiding:** Inheritance presents a feature to hiding data by marking a method as a private in the main class so that it cannot be utilized or alter by the sub-class.

In the object-oriented paradigm, the basis of inheritance is an "IS-A" bond, which describes "R is a Z type of thing", blue is a color, the bus is a vehicle. The inheritance is uni-directional, "the house is a building", but "the building is not a house" etc. The inheritance has other additional important characteristics (*Aziz, Khan & Nadeem, 2019*):

1. **Generalization:** dissemination of commonalities amongst several classes is termed as generalization (*Pason, 1994*).
2. **Specialization:** increasing the functionality of a class is described as specialization (*Breesam, 2007*).

The research shows that inheritance has various forms. There are described in the subsequent lines (*Shivam, 2013*):

1. **Single Inheritance:** when a sub-class only inherits through a single main-class is denoted as single inheritance.

**Table 1** SFP studies (1990–2003).

| Reference | Algorithms | Performance measure |
|---|---|---|
| *Porter & Selby (1990)* | Classification Tree | Accuracy |
| *Briand, Basili & Hetmanski (1993)* | Logistic Regression, Classification Tree, Optimized Set Reduction | Correctness, Completeness |
| *Lanubile, Lonigro & Visaggio (1995)* | PCA, Discriminant Analysis, Logistic Regression, Logical Classification | Misclassification Rate |
| *Cohen & Devanbu (1997)* | Foil, Flipper on IPL | Error Rate |
| *Khoshgoftaar et al. (1997)* | ANN and Discriminant | Type-I, Type-II, Misclassification Rate |
| *Evett et al. (1998)* | | Ordinal Evaluation Procedure |
| *Ohlsson, Zhao & Helander (1998)* | PCA, Discriminant Analysis for Classificaitn, Multivariate Analysis | Misclassification Rate |
| *Binkley & Schach (1998)* | Spearman Rank Correlation Test | |
| *De Almeida & Matwin (1999)* | C4.5, CN2, FOIL, NewID | Correctness, Completeness, Accuracy |
| *Kaszycki (1999)* | | TN, TP |
| *Yuan et al. (2000)* | Fuzzy Subtractive Clustering | Type-I, Type-II, Overall Misclassification Rate, Effectiveness, Efficiency |
| *Denaro (2000)* | Logistic Regression | R2 |
| *Khoshgoftaar, Gao & Szabo (2001)* | | Type-I, Type-II |
| *Xu, Khoshgoftaar & Allen (2000)* | PCA,FNR | Average Absolute Error |
| *Guo & Lyu (2000)* | Finite Mixture Model Analysis, Expectation Maximization (EM) | Type-II Error |
| *Khoshgoftaar, Gao & Szabo (2001)* | ZIP | AAE,ARE |
| *Schneidewind (2001)* | BDF,LRF | Type-I, Type-II, Misclassification Rate |
| *Emam, Melo & Machado (2001)* | Logistic Regression | J-Coefficient |
| *Khoshgoftaar, Geleyn & Gao (2002)* | PRM, ZIP, Module Order Modeling | Average Relative Error |
| *Khoshgoftaar (2002)* | GBDF | Type-I, Type-II |
| *Mahaweerawat, Sophasathit & Lursinsap (2002)* | Fuzzy Clustering, RBF | Type-I, Type-II, Misclassification Rate |
| *Khoshgoftaar & Seliya (2002a)* | SPRINT(Classification Tree),CART(Decision Tree) | Type-I,Type-II, Misclassification Rate |
| *Pizzi, Summers & Pedrycz (2002)* | Median-Adjusted Class Labels(Pre-Processing),Multilayer Perception | Accuracy |
| *Khoshgoftaar & Seliya (2002b)* | CART-LS,S-PLUS,CART-LAD | AAE,ARE |
| *Reformat (2003)* | Classification Models | Rate Change |
| *Koru & Tian (2003)* | Tree Base Models | U-Test |
| *Denaro, Lavazza & Pezzè (2003)* | Logistics Regression | |
| *Thwin & Quah (2003)* | GRNN | R2,R,ASE,AAE,Min AE, Max AE |
| *Khoshgoftaar & Seliya (2003)* | CART-LS,S-PLUS,CART-LAD | |
| *Guo, Cukic & Singh (2003)* | Dempster-Shafer Belief Networks | Probability of Detection, Accuracy |
| *Denaro, Pezzè & Morasca (2003)* | Logistic Regression | R2,Completeness |

2. **Multiple Inheritances:** in case of multiple inheritances, a sub-class is inheriting or expanding through many main-classes. The problem in multiple inheritance is the sub-class would manage the dependencies of many main-classes.

**Table 2  SFP studies (2004–2007).**

| Reference | Algorithms | Performance measure |
|---|---|---|
| *Menzies & Di Stefano (2004)* | Naïve Bayes, J48 | PF |
| *Khoshgoftaar & Seliya (2004)* | CART,S-PLUS,SPRING-Sliq,C4.5 | Misclassification Rate |
| *Wang, Yu & Zhu (2004)* | CGA,ANN | Accuracy |
| *Mahaweerawat et al. (2004)* | RBF | Accuracy, type-I, Type-II |
| *Menzies & Di Stefano (2004)* | LSR, Model Trees, ROCKY | Accuracy, Sensitivity, Precesion |
| *Kaminsky & Boetticher (2004)* | Genetic Algorithm | *T*-test |
| *Kanmani et al. (2004)* | GRNN,PCA | r,R2,ASE,AAE,Max AE, Min AE |
| *Zhong, Khoshgoftaar & Seliya (2004)* | K-means, Neural-Gas clustering | MSE, FPR, FNR, Misclassification Rate |
| *Xing, Guo & Lyu (2005)* | SVM, QDA, Classification Tree | Type-I, Type-II Error |
| *Koru & Liu (2005a)* | J48,Kstar,Bayesian Networks, ANN,SVM | F-measure |
| *Khoshgoftaar, Seliya & Gao (2005)* | C4.5,Decesion Tree, Discriminant Analysis, Logistic Regression | Misclassification Rate |
| *Koru & Liu (2005b)* | J48,K-Star,Random Forests | F-measure |
| *Challagulla et al. (2005)* | Linear Regression, SVM, Naïve Bayes,J48 | AAE |
| *Gyimothy, Ferenc & Siket (2005)* | Logistic Regression, Linear Regression, Decision trees, NN | Completeness, Correctness, Precision |
| *Ostrand, Weyuker & Bell (2005)* | Negative Binomial Regression Model | Accuracy |
| *Tomaszewski, Lundberg & Grahn (2005)* | Regression Technique, PCA | R2 |
| *Hassan & Holt (2005)* | | Hit Rate, APA |
| *Ma, Guo & Cukic (2006)* | Random Forests, LR, DA, Naïve Bayes, J48,ROCKY | G-mean I,G-mean II,F-measure, ROC, PD, Accuracy |
| *Challagulla, Bastani & Yen (2006)* | MBR | PD, Accuracy |
| *Khoshgoftaar, Seliya & Sundaresh (2006)* | MLR,CBR | ARE,AAE |
| *Nikora & Munson (2006)* | Rules for Fault Definition | |
| *Zhou & Leung (2006)* | Logistic Regression, Naïve Bayes, Random Forest | Correctness, Completeness, Precision |
| *Mertik et al. (2006)* | C4.5, SVM,RBF | PD,PF,Accuracy |
| *Gao & Khoshgoftaar (2007)* | Poisson Regression, Negative Bionomial Regression, Hardle Regression | AAE,ARE |
| *Li & Reformat (2007)* | SimBoost | Accuracy |
| *Mahaweerawat, Sophatsathit & Lursinsap (2007)* | RBP,Self-Organizing Map Clustering | MAR |
| *Menzies, Greenwald & Frank (2007)* | Naïve Bayes, J48 | PD, PF, Balance |
| *Ostrand, Weyuker & Bell (2007)* | Negative Binomial Regression Model | accuracy |

**Table 2** (*continued*)

| Reference | Algorithms | Performance measure |
|---|---|---|
| *Pai & Dugan (2007)* | Linear Regression, Poisson Regression, Logic Regression | Sensitivity, Specificity, Precision, FP, FN |
| *Wang, Zhu & Yu (2007)* | S-PLUS, TreeDisc | Type-I, Type-II |
| *Seliya & Khoshgoftaar (2007)* | EM Techniques | Type-I, Type-II |
| *Tomaszewski et al. (2007)* | Univariate Liner regression Analysis | Accuracy |
| *Seliya & Khoshgoftaar (2007)* | Semi-Supervised Clustering, K-means Clustering | Type-I,Type-II |
| *Olague, Gholston & Quat-tlebaum (2007)* | UBLR, Spearman Correlation | Accuracy |
| *Jiang, Cukic & Menzies (2007)* | Naïve Bayes, Logistic Regression, J48,IBK,Random Forests | PD,PF |

3. **Multilevel Inheritance:** multilevel inheritance in object-oriented paradigm, refers to an approach when a sub-class spread out from a derived class, making the derived class a main class of the freshly formed class.
4. **Hierarchical Inheritance:** in the situation of hierarchical inheritance one superclass is expanded by means of many sub-classes.
5. **Hybrid Inheritance:** is a mixture of multi-layer and multiple inheritance. In multiple inheritance, subclasses are expended from two superclasses. Although these superclasses are derived classes rather than the base classes.

## Inheritance metrics

**Depth of Inheritance Tree (DIT)** (*Chidamber & Kemerer, 1994*)**:** the DIT metrics is a measurement of how significantly subclasses may efface the metrics of this class. In the case of multiple inheritances, DIT would be the maximum distance from the node to the root of a tree.

$$DIT = Max\ inheritance\ path\ from\ the\ class\ to\ the\ root \tag{1}$$

- The lowest class in the hierarchy would inherit a larger amount of methods, consequently, it would be hard to predict their behavior.
- While in the design phase, deeper trees will create more complexity since several classes and methods are being used.
- The lower a particular class is in the hierarchy, the higher possibility reuse of inherited methods.

## Number of children (NOC) (*Chidamber & Kemerer, 1994*)

- Increasing the NOC will rise in reuse since inheritance is a type of reuse.
- The larger the sub-class, the bigger the probability of inadequate abstraction of the main-class. In the case where a class has a large number of subclasses, it would be a situation of misappropriation of a child-class.

**Table 3  SFP studies (2008–2020).**

| Reference | Algorithms | Performance Measure |
|---|---|---|
| *Bibi et al. (2008)* | RvC | AAE, Accuracy |
| *Bingbing et al. (2008)* | K-means, Affinity Propagation | Type-I, Type-II |
| *Marcus, Poshyvanyk & Ferenc (2008)* | Univariate Logistic Regression | Precision, Correctness |
| *Shafi et al. (2008)* | Classification Via Regression | Precision, Recall, Accuracy |
| *Catal & Diri (2009a)* | Random Forests(Artifical Immune Systems),Naïve Bayes | AUC |
| *Turhan, Kocak & Bener (2009)* | CBGR, Nearest Neighbor Sampling | |
| *Catal & Diri (2009b)* | X-means Clustering, | |
| *Catal, Sevim & Diri (2009)* | Naïve Bayes, YATSI | |
| *Alan & Catal (2009)* | | |
| *Suresh, Kumar & Rath (2014)* | Linear Regression, Logistic Regression, ANN | Precision, Correctness, Completeness, MAE,MARE,RMSE,SEM |
| *Aleem, Capretz & Ahmed (2015)* | NaiveBayes, MLP, SVM, AdaBoost, Bagging, Decision Tree, Random Forest, J48, KNN, RBF and K-means | Accuracy, Mean absolute error and F-measure |
| *Rathore & Kumar (2015)* | Genetic Programming(GP) | Error rate, Recall, Completeness |
| *Yohannese & Li (2017)* | NB, NN, SVM, RF, KNN, DTr, DTa, and RTr | ROC |
| *Pahal & Chillar (2017)* | ANN, SSO | Accuracy |
| *Mohapatra & Ray (2018)* | GSO-GA,SVM | Fitness Value, Accuracy |
| *Wójcicki & Dabrowski (2018)* | Machine Learning | Recall, False Positive Rate |
| *Patil, Rao & Bindu (2018)* | Linear Regression, FCM | Coefficients, Standard Errors And T-Values |
| *Arasteh (2018)* | Naive Bayes, ANN, SVM | Accuracy, Precision |
| *Akour et al. (2019)* | GA,SVM | Accuracy, Sd, Error Rate, Specificity, Precision, Recall, And F-Measure |
| *Balogun et al. (2019)* | RIPPER, Bayesian Network, Random Tree, and Logistic Model Tree | Area Under Curve (AUC) |
| *Rajkumar & Viji (2019)* | SVM,ANN,KNN | Accuracy, Sensitivity, Specificity, Precision |
| *Alsaeedi & Khan (2019)* | SVM, DS, RF | Accuracy, Precision, Recall, F-Score, ROC-AUC |
| *Rhmann et al. (2020)* | Random Forest, J48 | Precision, Recall |
| *Sharma & Chandra (2020)* | Factor Analysis (FA) | R2, Adjusted R2 |
| *Ahmed et al. (2020)* | SVM | Precision, Recall, Specificity, F 1 Measure, Accuracy |

- The quantity of subclasses presents an impression of the possible impact on the design of a class. In case when a class contains larger numbers of subclasses, it would require further testing of methods present within the class.

$$NOC = number\ of\ immediate\ sub-classes\ of\ a\ class \tag{2}$$

**Attribute Inheritance Factor (AIF)** (*Abreu & Carapuça, 1994*)**:** AIF is the ratio of the sum of all classes inherited attributes in the system to the all classes total number of available attributes. It is a metric at system-level which gauges the range of inherited attribute within

the system. The equation to calculated AIF is as under:

$$AIF = \frac{\sum Ai(Ci)}{\sum Aa(Ci)} \tag{3}$$

**Method Inheritance Factor** (MIF) (*Abreu & Carapuça, 1994*): MIF is the ratio of sum of all classes inheritance methods of the system with the total number of all classes presented methods. It is a metric at the system-level. It is proposed to maintain MIF in between 0.25 and 0.37. The equation to calculated MIF is under:

$$MIF = \frac{\sum Mi(Ci)}{\sum Ma(Ci)} \tag{4}$$

**Number Of Methods Inherited** (NMI) (*Lorenz & Kidd, 1994*): NMI metric calculates the total methods inherited by a sub-class.

**Number of Methods Overridden** (NMO): a larger value of NMO reveals a design issue, showing that these methods were overridden as a last-minute design. It is recommended that a sub-class should be a specialization of its main classes, which results in a brand-new distinctive name for the methods.

**Number of New Methods** (NNA): the usual anticipation of a sub-class is how to additionally specialize or add up objects of the main class. If there is not any method with a similar name in any superclass, the said method is defined as an additional method in the subclass.

**Inheritance Coupling** (*Li & Henry, 1993*): is an association between classes that facilitates to use of earlier defined objects, consist of variables and methods. Inheritance reduces class complexity by decreasing the number of methods of a single class, but then this becomes design and maintenance difficult. Inheritance improves reusability and efficiency when using current objects. Simultaneously, inheritance has led to complexities in testing and understanding software. This implies that inheritance coupling affects several quality attributes such as complexity, reusability, efficiency, maintainability, understandability, and testability.

**Number Of Inherited Methods** (NIM): is a simple metric that describes the extent to which a particular class may reuse. It calculates the number of methods a class may gain access to in its main class. The greater the inheritance of methods, the greater the reuse of a class will be via subclasses. Comparing this metric amongst the number of superclasses referred and the manner it is referred to methods not specified in the class could be exciting since it shows how much internal reuse occurred within the calling class and its superclass. It may be an inward call to an inward method, even though it is hard to measure it. Also, inheriting from larger superclasses could be a problem for the reason that only a subset of the behavior may be used/needed in subclasses. This is a limit of the single inheritance based on the object-oriented paradigm.

**Fan-In and Fan-Out Metric** (*Henry & Kafura, 1981*): Henry and Kafura first defined the Fan-In and Fan-Out metrics (*Henry & Kafura, 1981*). These are "module-level" metrics and expanded for the object-oriented paradigm. Assuming a class X, we note its Fan-In such as the number of classes, that make use of characteristics of class X. Likewise, the Fan-Out for a class X is the number of classes utilized by X.

Sheetz, Tegarden, and Monarchi originate a set of basic counts. Fundamental complexity or inter-module complexity (*Card & Glass, 1990*) has been recognized as an important part of the complexity of a structured system. Several researchers have utilized module-defined Fan in and Fanout (*Belady & Evangelisti, 1981*; *Card & Glass, 1990*; *Monarchi & Puhr, 1992*). Extending these ideas to variables in object-oriented systems seems appropriate and straightforward. The number of methods using variables (variable Fan-In) is very similar to the number of modules calling the module (Fan-In), and the number of objects accessed by the variable (variable Fan-Out) and the digital module called by the module (Fan-Out).
**Fan-Down:** Fan-Down is the number of objects below the object hierarchy (subclasses).
**Fan-Up:** Fan-Up is the number of objects above in hierarchy (superclasses).
**Class-To-Root Depth:** the maximum number of levels in the hierarchy that are above the class.
**Class-To-Leaf Depth:** the maximum number of levels in the hierarchy that are below the class.
**Measure of Functional Abstraction (MFA):** MFA is the share of the count of methods inherited by a class with the sum of methods of the class. Its range is from 0 to 1.
**IFANIN:** IFANIN metric counts the immediate base classes in the hierarchy.

## Inheritance metrics and their usage

Inheritance is a key characteristic of the object-oriented paradigm. It facilitates the class level design and forms the "IS-A" relationship among classes since the basic segment of the development of a system is the design of class(*Rajnish, Choudhary & Agrawal, 2010*). The utilization of inheritance shrinkages the costs of testing efforts and maintenance of the system (*Chidamber & Kemerer, 1994*). The reuse employing inheritance will thus deliver software, that is greatly understandable, maintainable, and reliable(*Basili, Briand & Melo, 1996*). In an experiment, Harrison et al. describe the absence of inheritance as easier to control or grasp as compared to the software that makes use of inheritance aspect (*Harrison, Counsell & Nithi, 1998*). However, Daley's experiments reveal software with tertiary inheritance may possibly be simply revised as compared to the software with no inheritance aspect (*Daly et al., 1996*).

The Inheritance metrics calculate numerous aspects of inheritance, that include breadth and depth of the hierarchy, besides the overriding complexity (*Krishna & Joshi, 2010*). Similarly, Bhattacherjee and Rajnish executed a study about inheritance metrics related with classes (*Rajnish, Bhattacherjee & Singh, 2007*; *Rajnish & Bhattacherjee, 2008a*; *Rajnish & Bhattacherjee, 2007*; *Rajnish & Bhattacherjee, 2006a*; *Rajnish & Bhattacherjee, 2006b*; *Rajnish & Bhattacherjee, 2005*). However, it is agreed the deeper the inheritance hierarchy, the class reusability will be enhanced but system maintainability will be complicated. In order to streamline the insight, software designers striving to keep inheritance hierarchy narrow and dispose of reusability by the usage of inheritance (*Chidamber & Kemerer, 1994*). Hence, it's important to assess the difficulty of the inheritance hierarchy to resolve the disparity among depth and shallowness.

Several metrics focus on inheritance are well-defined by the researchers. These metrics with their references are listed in Table 4 (*Aziz, Khan & Nadeem, 2019*).

Some of these Inheritance metrics listed in Table 4 are discussed briefly in 'Inheritance metrics' of the paper. We specifically take these inheritance metrics in the background of object oriented software fault prediction.

### Data sets in SFP

In software fault prediction, so many data sets are being used. These data sets are categorized into public, private, partial, and unknown data sets types (*Catal & Diri, 2009a*). Out of these, public type data sets utilization has increased from 31% to 52% since 2005 onward (*Malhotra, 2015*). It is a fact that fault information normally not available for private projects however there are public data sets widely available with fault information, these can be downloaded for free. Also, there are many fault repositories, out of this Tera-PROMISE (*Boetticher, 2007*) warehouses, and D'Ambros warehouses are usually utilized for fault predicting (*D'Ambros, Lanza & Robbes, 2010*).

A publicly available repository identified as Tera-PROMISE presents substantial data sets of many projects. Its previous edition was named as NASA repository (*Shirabad & Menzies, 2005*). The data sets of NASA are a vital resource of the Tera-PROMISE repository because its data sets are a widely used library of SFP. Nearly 60% of articles published between 1991 and 2013 take advantage of this archive (*Card & Agresti, 1988*). The library of Tera-PROMISE presents metrics related to product and process along with digital and nominal class labels for buildup regression and classification models.

The D'Ambros library retains data sets of six software systems. These are Equinox, Eclipse JDT Core, Eclipse PDE UI, Framework, and Mylyn.

During the review of the literature, it is identified that scholars make use of private, and public data sets for the proof of their study. In this respect, Table 5 indicates the name of the author, publication year, and public data sets or private data sets applied in said experiments.

## LITERATURE REVIEW

In this section, emphases are on inheritance metrics to find out how these might be effective in SFP. This paper is not performing a systematic literature review. It explores how various inheritance metrics are advantageous in fault prediction.

### Inheritance in SFP

Object-oriented metrics are employed for the prediction of faults to produce quality software. The attributes that ascertain software quality are fault tolerance, understandability, defect density, maintainability, normalized rework rate, reusability, and many more others.

There are several metrics levels comprising class level, method level, file level, process level, component level, and quantitative levels. The method-level metrics are comprehensively applied for the prediction of faults problem. *Halstead (1977)* and *McCabe (1976)* metrics suggested in the year 1970s though these are still the greatest predominant metrics at the method-level. The class-level metrics are merely applied in object-oriented programs. The C&K (*Chidamber & Kemerer, 1994*) set of metrics is yet the ultimate predominant metrics suite at class-level being employed for fault prediction. Table 6

**Table 4  Inheritance metrics.**

| Author name | Metrics name |
| --- | --- |
| Chidamber and Kemerer (*Chidamber & Kemerer, 1994*) | Depth of Inheritance Tree (DIT) |
| Abreu Mood metrics suit (*Abreu & Carapuça, 1994*) | Number of Children (NOC) |
| | Attribute Inheritance Factor (AIF) |
| | Method Inheritance Factor (MIF) |
| Bansiya J. et al. QMOOD (*Bansiya & Davis, 2002*) | Number of Hierarchies (NOH) |
| | Average number of Ancestors (ANA) |
| | Measure of Functional Abstraction (MFA) |
| Henry's & Kafura (*Henry & Kafura, 1981*) | Fan in |
| | Fan out |
| Tang, Kao and Chen, (*Li & Henry, 1993*) | inheritance coupling(IC) |
| Lorenz and Kidd (*Lorenz & Kidd, 1994*) | Number of Method Inherited (NMI) |
| | Number of Methods Overridden (NMO) |
| | Number of New Methods(NNA) |
| | Number of Variable Inherited (NVI) |
| Henderson-Sellers (*Henderson-Sellers, 1995*) | AID (average inheritance depth) |
| Li (*Li, 1998*) | NAC (number of ancestor classes) |
| | NDC (number of descendent classes) |
| Tegarden et al. (*Tegarden, Sheetz & Monarchi, 1995*) | CLD (class-to-leaf depth) |
| | NOA (number of ancestor) |
| Lake and Cook (*Lake & Cook, 1994*) | NOP (number of parents) |
| | NOD (number of descendants) |
| Rajnish et al. (*Rajnish & Singh, 2013*; *Rajnish & Bhattacherjee, 2008*) | DITC (Depth of Inheritance Tree of a Class) |
| | CIT (Class Inheritance Tree) |
| Sandip et al. (*Catal, 2011*; *Mal & Rajnish, 2013*) | ICC (Inheritance Complexity of Class) |
| | ICT (Inheritance Complexity of Tree) |
| Gulia, Preeti, and Rajender S. Chillar (*Gulia & Chillar, 2012*) | CCDIT (Class Complexity Due To Depth of Inheritance Tree) |
| | CCNOC (Class Complexity Due To Number of Children) |
| F. T. Sheldon et al. (*Sheldon, Jerath & Chung, 2002*) | Average Degree of Understandability (AU) |
| | Average Degree of Modifiability (AM) |
| Rajnish and Choudhary (*Rajnish, Choudhary & Agrawal, 2010*) | Derive Base Ratio Metric (DBRM) |
| | Average Number of Direct Child (ANDC) |
| | Average Number of Indirect Child (ANIC) |
| Mishra, Deepti, and Alok Mishra (*Mishra & Mishra, 2011*) | CCI (Class Complexity due to Inheritance) |
| | ACI (Average Complexity of a program due to Inheritance) |
| | MC (Method Complexity) |
| | Total Children Count (TCC) |
| | Total Progeny Count (TPC) |
| Abreu and Carapuc (*Mishra & Mishra, 2011*; *e Abreu & Carapuça, 1994*) | Total Parent Count (TPAC) |
| | Total Ascendancy Count(TAC) |
| | Total Length of Inheritance chain (TLI) |
| | Method Inheritance Factor(MIF) |

*(continued on next page)*

**Table 4** (*continued*)

| Author name | Metrics name |
|---|---|
| K. Rajnish and A. K. Choudhary (*Rajnish, Choudhary & Agrawal, 2010*) | Extended Derived Base Ratio Metrics (EDBRM) |
| | Extended Average Number of Direct Child (EANDC) |
| | Extended Average Number of Indirect Child (EANIC) |
| Rajnish and Bhattacherjee (*Rajnish & Bhattacherjee, 2007*) | Inheritance Metric Tree (IMT) |
| Chen, J. Y., and J. F. Lu (*Chen & Lu, 1993*) | Class Hierarchy of Method (CHM) |
| Lee et al. (*Lee, 1995*) | Information-flow-based inheritance coupling (IH-ICP) |

recapped the commonly employed metrics set at class, method, and file-level for software fault prediction.

Many studies have been carried out on SFP which also takes into account the object-oriented metrics. These studies includes, empirical study on open source software for fault prediction using `{loc}`, `{ dit }`, `{ noc }`, `{ lcom }` and `{ cbo }` metrics (*Gyimothy, Ferenc & Siket, 2005*). The reusability study on object-oriented software using inheritance, cohesion, and coupling metrics (*Catal, 2012*). The experimental-based assessment of C&K metrics (*Kumar & Gupta, 2012*), reusability metrics for object-oriented design (*Goel & Bhatia, 2012*), and empirical analysis of C&K metrics for the object-oriented design complexity (*Subramanyam & Krishnan, 2003*).

The metrics collection of C&K crafted and applied by *Chidamber & Kemerer (1994)* are the ultimate frequently utilize metrics set for software related to object oriented. *Briand et al. (2000)* have analyzed the collection of object oriented design metrics suggested by *Basili, Briand & Melo (1996)*. R. Subramanyam validated that `{dit}`, `{cbo}` and `{wmc}` metrics are fault predictor at class level (*Subramanyam & Krishnan, 2003*).

Experimental evaluations of the classification algorithm have been built for fault prediction through researches (*Kaur & Kaur, 2018*). Basili et al. revealed that many C&K metrics are observed to be associated with failure propensity (*Basili, Briand & Melo, 1996*). Tang et al. assessed C&K metrics suite and discovered that any of these metrics except `{rfc}` and `{wmc}` were deemed vital (*Tang, Kao & Chen, 1999*). Briand et al. carryout forty-nine metrics to ascertain, which model to apply for the prediction of faults. Conclusions shows apart from `{noc}` all metrics are useful to predict faults tendency (*Briand et al., 2000*). Wust and Briand determined that `{dit}` metrics are inversely correlated to fault proneness and `{noc}` metrics is an insignificant predictor of fault tendency (*Briand, Wüst & Lounis, 2001*). Yu et al. selected eight metrics to explored the relationship amongst these metrics and the tendency to identify faults. So firstly they explored the correlation among metrics and found four closely associated sets. After this, they utilize univariate analysis to observe, which set can classify faults (*Yu, Systa & Muller, 2002*). Malhotra and Jain applied logistic regression methods to examine the correlation amongst metrics of object-oriented along with faults tendency. The receiver operating characteristics (ROC) evaluation was employed. The predictive model performance was assessed through ROC (*Malhotra & Jain, 2012*). Yeresime et al. have investigated using linear regression, logistic regression, and artificial neural network methods for the prediction of software faults making use of

**Table 5  Data usage by studies.**

| Author | Year | Dataset |
|---|---|---|
| Briand et al. | 2000 | Hypothetical video rental business |
| Cartwright et al. | 2000 | Large European telecommunication industry, which consists of 32 classes and 133KLOC. |
| Emam et al. | 2001 | Used two versions of Java application: Ver 0.5 and Ver 0.6 consisting of 69 and 42 classes. |
| Gyimothy et al. | 2005 | Source code of Mozilla with the use of Columbus framework |
| Nachiappan et al. | 2005 | Open source eclipse plug-in |
| Zhou et al. | 2006 | NASA consisting of 145 classes, 2107 methods and 40 KLOC |
| Olague H.M et al. | 2007 | Mozilla Rhino project |
| Kanmani et al. | 2007 | Library management system consists of 1185 classes |
| Pai et al. | 2007 | Public domain dataset consists of 2107 methods, 145 classes, and 43 KLOC |
| Tomaszewksi et al. | 2007 | Two telecommunication project developed by Ericsson |
| Shatnawi et al. | 2008 | Eclipse project: Bugzilla database and Change log |
| Aggarwal et al. | 2009 | Student projects at University School of Information Technology |
| Singh et al. | 2009 | NASA consists of 145 classes, 2107 methods and 40K LOC |
| Cruz et al. | 2009 | 638 classes of Mylyn software |
| Burrows et al. | 2010 | iBATIS, Health watcher, Mobile media |
| Singh et al. | 2010 | NASA consists of 145 classes, 2107 methods, and 40K LOC |
| Zhou et al. | 2010 | Three releases of Eclipse, consisting of 6751, 7909, 10635 java classes and 796, 988, 1306 KLOC |
| Fokaefs et al. | 2011 | NASA datasets |
| Malhotra et al. | 2011 | Open source software |
| Mishra et al. | 2012 | Eclipse and Equinox datasets |
| Malhotra et al. | 2012 | Apache POI |
| Heena | 2013 | Open Source Eclipse System |
| Rinkaj Goyal et al. | 2014 | Eclipse, Mylyn, Equinox and PDE |
| Yeresime et al. | 2014 | Apache integration framework (AIF) Ver 1.6 |
| Ezgi Erturk et al. | 2015 | Promise software engineering repository data |
| Golnoush Abaei et al. | 2015 | NASA datasets |
| Saiqa Aleem et al. | 2015 | PROMISE data repository |
| Santosh et al. | 2015 | 10 Datasets from PROMISE repository |
| Yohannese et al. | 2016 | AEEEM Datasets & Four datasets PROMISE repository |
| Ankit Pahal et al. | 2017 | Four projects from the NASA repository |
| Bartłomiej et al. | 2018 | Github Projects: Flask, Odoo, GitPython, Ansible,Grab |
| Patil et al. | 2018 | Real-time data set, Attitude Survey Data |
| Bahman et al. | 2018 | Five NASA datasets |
| Hiba Alsghaier et al. | 2019 | 12-NASA MDP and 12-Java open-source projects |
| Balogun et al. | 2019 | NASA and PROMISE repositories |
| Alsaeedi et al. | 2019 | 10 NASA datasets |
| Wasiur Rhmann et al. | 2020 | GIT repository, Android-4 & 5 versions |
| Deepak et al. | 2020 | Open source bug metris dataset |
| Razu et al. | 2020 | 3 open source datasets from PROMISE |

C&K metrics. The findings show the significance of the weighted method per class {wmc} metric for classification of fault (*Suresh, Kumar & Rath, 2014*). The impact of inheritance metrics in SFP is authenticated in experimentation, where the artificial neural network

**Table 6  Frequently used metrics in software fault prediction.**

**Method level metrics**

1. loc McCab's line count of code

2. v(g) McCabe "cyclomatic complexity"

3. ev(g) McCabe "essential complexity"

4. iv(g) McCabe "design complexity"

5. n Halstead total operators + operands

6. v Halstead "volume"

7. l Halstead "program length"

8. d Halstead "difficulty"

9. i Halstead "intelligence"

10. e Halstead "effort"

11. b Halstead "bug"

12. t Halstead's time estimator

13. lOCode Halstead's line count

14. lOComment Halstead's count of lines of comments

15. lOBlank Halstead's count of blank lines

16. lOCodeAndComment Lines of comment and code

17. uniq_op Halstead Unique operators

18. uniq_opnd Halstead Unique operands

19. total_op Halstead Total operators

20. total_opnd Halstead Total operands

21. branchCount Branch count of the flow graph

Converted method-level metrics into class-level using minimum , maximum , average and sum operations (21*4=84)

**Class Level Metrics**

1. Coupling between Objects (CBO)

2. Depth of Inheritance Tree (DIT)

3. Lack of Cohesion of Methods (LCOM)

4. Number of Children (NOC)

5. Response for Class (RFC)

6. Weighted Method Per Class (WMC)

Percent_Pub_Data

Access_To_Pub_Data

Dep_On_Child

Fan_In is the # of calls by higher modules.

No_of_Method

No_Of_Attribute

No_Of_Attribute_Inher

No_Of_Method_Inher

**Table 6** (*continued*)

| Class Level Metrics |
| --- |
| Fan-in |
| Fan_out |
| No_of_Pvt_Method |
| No_of_Pvt_Attribute |
| No_Pub_Method |
| No_Pub_Attribute |
| NLOC |

| File Leel Metrics |
| --- |
| 1. # of times the source file was inspected prior to system test release. |
| 2. # of LOC for the source file prior to coding phase (auto-generated code) |
| 3. # of LOC for the source file prior to system test release. |
| 4. # of lines of commented code for the source prior to code(auto-generated). |
| 5. # of lines of commented code for the source file prior to system test release. |

(ANN) is utilized for model building. Recall, Accuracy, F1 measures, True negative rate (TNR), and Precision are employed for performance measures. The conclusions reveal the acceptable impact of inheritance metrics SFP. The practitioners of software testing can strongly use inheritance metrics to predict faults in software projects. A higher value of inheritance is also detrimental, because subsequently, it may generate faults in software (*Aziz, Khan & Nadeem, 2019*).

The literature review shows that the `{ dit }` and `{ noc }` metrics, related to inheritance feature are utilized in the prediction of faults jointly inside C&K metrics suite only. Consequently, it is considered important to validate the exclusive value of inheritance metrics in the context of software fault prediction.

## METHODOLOGY

The research approach comprises on three interconnected phases, as shown in Fig. 1, which includes selection, preprocessing, and experimentation/evaluation phases. The first phase comprises on the choice of data sets of inheritance metrics and performance measure. The selection of inheritance metrics data sets is based on dual criteria, the data set should be publicly available and metrics correlation should not be $\geq 0.7$ OR $\leq -0.7$.

In the second pre-processing phase, other metrics in addition to the inheritance for example `loc`, `cbo`, `wmc` are removed to keep the data set consistent. These data sets are then split by all possible combinations of their features sets. After this, the data sets are cleaned, filtered, and remove the related anomalies. Finally, in the experiment/evaluation phase, the final form of the data set is used for the experiment, in which SVM (Support Vector Machine) is built and cross-validated. The calculation of cross entropy losses, Accuracy, F-Measure and

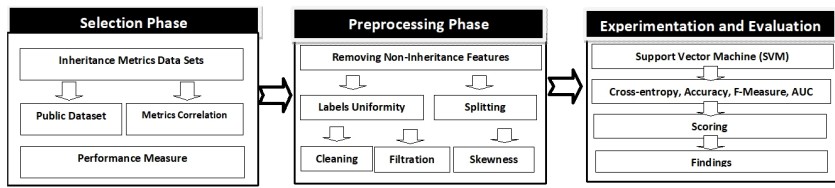

**Figure 1  Research methodology.**

AUC is performed. Accordingly, a score is calculated for each selected inheritance metric to determine the superior.

## Selection phase

**(1) Selection of inheritance metrics:** from the inheritance metrics mentioned in 'Theoretical Background' of this paper, we choose only those metrics that meet the criteria stated as follows.

### Data set must be publicly available

This condition is comprehended since the software projects fault information is extremely rarely approachable. The fundamental issue is that the information of fault for large enterprise projects is stored digitally and propriety. The bug information on small projects is too less but available for the public. Thus, labeled data is infrequently accessible. The accessibility of the data set, which is publicly available will permit the assessment of the metrics related to inheritance in fault prediction. Lastly, 40 data sets were discovered with inheritance metrics (*Jureczko & Madeyski, 2010*; *Menzies & Di Stefano, 2004*; *Menzies et al., 2004*; *a51, 2009*; *Niu & Mahmoud, 2012*; *D'Ambros, Lanza & Robbes, 2010*; *Wagner, 2010*; *Abdelmoez, Goseva-Popstojanova & Ammar, 2006*; *Abdelmoez et al., 2005*; *Monarchi & Puhr, 1992*; *Shepperd et al., 2013*). A total of nine inheritance metrics are found in this data set; which are Inheritance Tree Depth (dit), Number of Children (noc), Functional Abstraction Measure (mfa), Inheritance Coupling (ic), Number of Method Inherited (nomi), Inherited Attribute Number (noai), Dependent on Child (doc) , Number of methods called per class (fanOut) and Number of classes that call class methods (fanIn).

A total of 40 data sets, out of which about 35 data sets are found on the servers of tera-PROMISE repository (*Boetticher, 2007*) and five data sets are located in the D'Ambros repository (*D'Ambros, Lanza & Robbes, 2010*). In this regard, Table 7 depicts the detailed information of these data sets. The first column indicates the data set name along with the version number if exist. Second column shows the detail about the total number of records and third column shows the percentage of fault for each base data sets. Overall, nine distinct metrics of inheritance are discovered in 40 data sets, where 3is used to label the presence of a metric in the associated data set and × is labeled where the metrics are not present in the data set.

Unluckily, all nine inheritance metrics do not exist in a single data set. However, set of inheritance metrics comprising {dit, noc, ic, mfa} are found in 30 data sets, {dit,

**Table 7  Source data-sets.**

| Dataset name | # Ins | % Falty | dit | noc | ic | mfa | noai | nomi | doc | fanin | fanout |
|---|---|---|---|---|---|---|---|---|---|---|---|
| ant-1.7 | 745 | 22 | ✓ | ✓ | ✓ | ✓ | × | × | × | × | × |
| Arc | 234 | 11 | ✓ | ✓ | ✓ | ✓ | × | × | × | × | × |
| berek | 43 | 37 | ✓ | ✓ | ✓ | ✓ | × | × | × | × | × |
| camel-1.2 | 608 | 36 | ✓ | ✓ | ✓ | ✓ | × | × | × | × | × |
| churn | 997 | 21 | ✓ | ✓ | × | × | ✓ | ✓ | × | ✓ | ✓ |
| ckjm | 10 | 50 | ✓ | ✓ | ✓ | ✓ | × | × | × | × | × |
| Eclipse JDT Core | 997 | 21 | ✓ | ✓ | × | × | ✓ | ✓ | × | ✓ | ✓ |
| Eclipse PDE UI | 1497 | 14 | ✓ | ✓ | × | × | ✓ | ✓ | × | ✓ | ✓ |
| eclipse34_debug | 1065 | 25 | ✓ | ✓ | × | × | ✓ | ✓ | × | × | × |
| eclipse34_swt | 1485 | 44 | ✓ | ✓ | × | × | ✓ | ✓ | × | × | × |
| e-learning | 64 | 9 | ✓ | ✓ | ✓ | ✓ | × | × | × | × | × |
| Equinox Framework | 324 | 40 | ✓ | ✓ | × | × | ✓ | ✓ | × | ✓ | ✓ |
| forrest-0.6 | 7 | 14 | ✓ | ✓ | ✓ | ✓ | × | × | × | × | × |
| iny-1.1 | 111 | 57 | ✓ | ✓ | ✓ | ✓ | × | × | × | × | × |
| jedit-3.2 | 272 | 33 | ✓ | ✓ | ✓ | ✓ | × | × | × | × | × |
| Kalkulator | 27 | 22 | ✓ | ✓ | ✓ | ✓ | × | × | × | × | × |
| Kc1-class-binary | 145 | 41 | ✓ | ✓ | ✓ | × | × | × | ✓ | ✓ | × |
| log4j-1.0 | 135 | 25 | ✓ | ✓ | ✓ | ✓ | × | × | × | × | × |
| Lucene | 691 | 9 | ✓ | ✓ | × | × | ✓ | ✓ | × | ✓ | ✓ |
| mylyn | 1862 | 13 | ✓ | ✓ | × | × | ✓ | ✓ | × | ✓ | ✓ |
| nieruchomosci | 27 | 37 | ✓ | ✓ | ✓ | ✓ | × | × | × | × | × |
| pdftranslator | 33 | 45 | ✓ | ✓ | ✓ | ✓ | × | × | × | × | × |
| poi-1.5 | 237 | 59 | ✓ | ✓ | ✓ | ✓ | × | × | × | × | × |
| prop-1 | 18471 | 15 | ✓ | ✓ | ✓ | ✓ | × | × | × | × | × |
| redaktor | 176 | 15 | ✓ | ✓ | ✓ | ✓ | × | × | × | × | × |
| serapion | 45 | 20 | ✓ | ✓ | ✓ | ✓ | × | × | × | × | × |
| single-version-ck-oo | 997 | 20 | ✓ | ✓ | × | × | ✓ | ✓ | × | ✓ | ✓ |
| skarbonka | 45 | 20 | ✓ | ✓ | ✓ | ✓ | × | × | × | × | × |
| sklebagd | 20 | 60 | ✓ | ✓ | ✓ | ✓ | × | × | × | × | × |
| synapse-1.0 | 157 | 10 | ✓ | ✓ | ✓ | ✓ | × | × | × | × | × |
| systemdata | 65 | 13 | ✓ | ✓ | ✓ | ✓ | × | × | × | × | × |
| szybkafucha | 25 | 56 | ✓ | ✓ | ✓ | ✓ | × | × | × | × | × |
| tempoproject | 42 | 30 | ✓ | ✓ | ✓ | ✓ | × | × | × | × | × |
| tomcat | 858 | 8 | ✓ | ✓ | ✓ | ✓ | × | × | × | × | × |
| velocity-1.4 | 196 | 75 | ✓ | ✓ | ✓ | ✓ | × | × | × | × | × |
| workflow | 39 | 51 | ✓ | ✓ | ✓ | ✓ | × | × | × | × | × |
| wspomaganiepi | 18 | 67 | ✓ | ✓ | ✓ | ✓ | × | × | × | × | × |
| xalan-2.4 | 723 | 15 | ✓ | ✓ | ✓ | ✓ | × | × | × | × | × |
| xerces-init | 162 | 47 | ✓ | ✓ | ✓ | ✓ | × | × | × | × | × |
| zuzel | 29 | 44 | ✓ | ✓ | ✓ | ✓ | × | × | × | × | × |

noc, nomi,noai} in 2 data sets, {dit, fanin, fanout, noai, noc, nomi} in 7 data sets and {dit, noc,ic, doc, fanin} in 1 data set.

Overall {dit} and {noc} features exist in all forty data sets. The feature {ic} is found in 31 data sets, feature mfa} in 30 data sets and similarly other in multiple data sets through these 659 data sets created for the experiment that will be explained in subsequent sections. The conclusion drawn is based upon the predictive ability of {ic} on 31 data sets. Same goes for rest of the feature sets.

It is prominent to mentioned here that some of the data sets are already utilized in an experiment to comparing inheritance metrics with C&K metrics (*Aziz, Khan & Nadeem, 2019*).

### Correlation should not be ≥ 0.7 OR ≤ −0.7

Software metrics have a tendency of correlation as these focus on the related characteristic of object-oriented programming for example in our case inheritance. A high value of correlation as $\geq 0.7$ or $\leq -0.7$ is a category of repetition that needs that repetition metric should be eliminated. The issue is the effect of managing the repetition metric might be negative, triggering uncertainty for the mining algorithm and determine a depleted pattern of quality (*Han, Pei & Kamber, 2011*). Furthermore, the advantages of removing correlated metrics are significantly better as compare to the cost (*Jiarpakdee, Tantithamthavorn & Hassan, 2018*). In the event of a lesser value of correlation, almost near to $\geq 0.7$ or $\leq -0.7$, the rejection of a metric may deprive the data set of significant important information.

In the study, in the case metrics illustrated in the second criterion, we execute the Pearson $r$ and the Spearman $p$ correlation coefficient for the pairs discovered in forty selected public data sets. Correlation between the features of each data set is computed in unfiltered data set. In order to explain the feature to feature correlation analysis further data set ant−1.7 has four features {dit}, {noc}, {ic}, and {mfa} as shown in Table 7. First the correlation of {dit} with other available features {noc}, {ic}, and {mfa} is computed. After this, the correlation of second feature of ant−1.7 data set {noc} with other features {ic} and {mfa} is computed. Finally {ic} with {mfa} is computed. The same process is separately applied between the features of all selected 40 data set shown in Table 7.

The strongly correlated features are dropped in their corresponding data sets only. It is important to mention here that correlation is only a precautionary step otherwise it does not manipulate feature set.

The presence of strongly correlated features in the SVM models make it difficult to converge and justify the generality of the results. Therefore, strongly correlated features are identified and dropped. Pearson correlation coefficient is used for the purpose.

Nevertheless, all combinations are positively correlated, where not a single pair is equal to $\geq 0.7$ or $\leq -0.7$. The nine inheritance metrics also meet the second criterion.

Tools/programming language and environments used to compute Pearson $r$ and the Spearman $p$ correlation coefficient is R platform with ggp4br R package.

2) **Selection of Performance Measure.** The models of machine learning built using classification are measured with their accomplishment by categorizing the unidentified

occurrences. A confusion matrix is a method of showing its capability. Catal et al. have computed many performance metrics, derived and originated through confusion matrix (*Catal, 2012*). Malhotra also has suggested the overall explanation of many assessment measures used in software fault prediction. The finding revealed that True Positive Rate(TPR) is the frequently utilized performance measure in software fault prediction, succeeding measures are Precision and AUC (*Malhotra, 2015*).

Cross entropy is a mean to compute the overall deviation of the models' probability from the actual label. Independent of the threshold is a key property of cross entropy (*Hinojosa et al., 2018*). It is effective in both training and testing phases (*Golik, Doetsch & Ney, 2013*; *Kline & Berardi, 2005*). So cross entropy is choose as a primary performance measure and Accuracy, F-Measure and AUC are selected as a supporting measure for this experiment.

## Preprocessing phase

**(1) Remove non-inheritance metrics.** Collected data sets have numerous metrics other than inheritance metrics, including {loc}, {wmc}, {ca}, {cbo}, and several others. Because we are planning to assess the inheritance metrics in the background of SFP, so all non-inheritance metrics are removed. This might affect the performance but it will be better to evaluate the inheritance metrics viability on software fault prediction.

**(2) Uniformity of Labels.** Every metrics encompass continuous numeric values with the associated data set however discrepancies are located in the class namely [bug]. These are settled through the guidelines mentioned as under:

$$bug = \begin{Bmatrix} \text{False} & \text{defects} = \text{No, False, 0, N, No} \\ & \text{True Otherwise} \end{Bmatrix} \tag{5}$$

Within the guidelines, [False] is employed when there are no faults and [True] represents faulty instances.

**3) Splitting.** The main objective of this study is to measure the significance of nine selected metrics of inheritance so forty data sets are separated into numerous possible sets of features after removing non-inheritance metrics.

The objectives of splitting process are to visualize the impact of every possible feature set and determine the most significant feature set out of the available data sets.

Each data set out of these forty data sets are separated into numerous possible sets of features by splitting and combining features into all possible unique combinations. In order to explain further, data set ant−1.7 has four features {dit}, {noc}, {ic}, and {mfa} as shown in Table 7. Splitting this single data set into all possible unique combinations will create about 15 sub-data sets as explained by the formula $2^{[number\ of\ features]} - 1$ where $2^4 - 1 = 15$ unique sub-data sets.

Resultantly 659 sub-data sets are formed by applying the same process on all forty publicly acquired data sets. Overall 67 unique features has been generated from these sub-data sets. First column of Table 8 shows these 67 unique features under the heading "features", 2nd column depicts the number of metrics combined to generate a unique feature and third column shows total number of sub-data sets formed by applying splitting process on to forty data sets that generated overall 659 sub-data sets.

**Table 8  Filtered data.**

| Features | # F | Total datasets | Small datasets | skewed | Remaining |
|---|---|---|---|---|---|
| mfa | 1 | 28 | 25 | 0 | 3 |
| noai | 1 | 9 | 7 | 0 | 2 |
| nomi | 1 | 9 | 5 | 1 | 3 |
| dit,fanIn | 2 | 9 | 5 | 0 | 4 |
| dit,fanOut | 2 | 7 | 3 | 0 | 4 |
| dit,mfa | 2 | 28 | 25 | 0 | 3 |
| dit,noai | 2 | 9 | 4 | 0 | 5 |
| dit,noc | 2 | 40 | 38 | 0 | 2 |
| dit,nomi | 2 | 9 | 3 | 0 | 6 |
| fanIn,fanOut | 2 | 6 | 0 | 0 | 6 |
| fanIn,noai | 2 | 7 | 1 | 0 | 6 |
| fanIn,noc | 2 | 8 | 4 | 0 | 4 |
| fanIn,nomi | 2 | 6 | 0 | 0 | 6 |
| fanOut,noai | 2 | 7 | 2 | 0 | 5 |
| fanOut,noc | 2 | 7 | 3 | 0 | 4 |
| fanOut,nomi | 2 | 6 | 0 | 0 | 6 |
| ic,mfa | 2 | 27 | 24 | 0 | 3 |
| noai,nomi | 2 | 9 | 1 | 0 | 8 |
| noc,mfa | 2 | 28 | 25 | 0 | 3 |
| noc,noai | 2 | 9 | 4 | 0 | 5 |
| noc,nomi | 2 | 9 | 2 | 0 | 7 |
| dit,fanIn,fanOut | 3 | 6 | 0 | 0 | 6 |
| dit,fanIn,noai | 3 | 7 | 1 | 0 | 6 |
| dit,fanIn,noc | 3 | 8 | 3 | 0 | 5 |
| dit,fanIn,nomi | 3 | 6 | 0 | 0 | 6 |
| dit,fanOut,noai | 3 | 7 | 1 | 0 | 6 |
| dit,fanOut,noc | 3 | 7 | 2 | 0 | 5 |
| dit,fanOut,nomi | 3 | 6 | 0 | 0 | 6 |
| dit,ic,mfa | 3 | 28 | 24 | 0 | 4 |
| dit,noai,nomi | 3 | 9 | 1 | 0 | 8 |
| dit,mfa,noc | 3 | 27 | 24 | 0 | 3 |
| dit,noc,noai | 3 | 9 | 2 | 0 | 7 |
| dit,noc,nomi | 3 | 9 | 1 | 0 | 8 |
| fanIn,fanOut,noai | 3 | 6 | 0 | 0 | 6 |
| fanIn,fanOut,noc | 3 | 6 | 0 | 0 | 6 |
| fanIn,fanOut,nomi | 3 | 6 | 0 | 0 | 6 |
| fanIn,noai,nomi | 3 | 6 | 0 | 0 | 6 |
| fanIn,noc,noai | 3 | 7 | 1 | 0 | 6 |
| fanIn,noc,nomi | 3 | 6 | 0 | 0 | 6 |

**Table 8** (*continued*)

| Features | # F | Total datasets | Small datasets | skewed | Remaining |
|----------|-----|----------------|----------------|--------|-----------|
| fanOut,noai,nomi | 3 | 6 | 0 | 0 | 6 |
| fanOut,noc,noai | 3 | 7 | 1 | 0 | 6 |
| fanOut,noc,nomi | 3 | 6 | 0 | 0 | 6 |
| ic,mfa,noc | 3 | 27 | 24 | 0 | 3 |
| noc,noai,nomi | 3 | 9 | 1 | 0 | 8 |
| dit,fanIn,fanOut,noai | 4 | 6 | 0 | 0 | 6 |
| dit,fanIn,fanOut,noc | 4 | 6 | 0 | 0 | 6 |
| dit,fanIn,fanOut,nomi | 4 | 6 | 0 | 0 | 6 |
| dit,fanIn,noai,nomi | 4 | 6 | 0 | 0 | 6 |
| dit,fanIn,noc,noai | 4 | 7 | 1 | 0 | 6 |
| dit,fanIn,noc,nomi | 4 | 6 | 0 | 0 | 6 |
| dit,fanOut,noai,nomi | 4 | 6 | 0 | 0 | 6 |
| dit,fanOut,noc,noai | 4 | 6 | 0 | 0 | 6 |
| dit,fanOut,noc,nomi | 4 | 6 | 0 | 0 | 6 |
| dit,ic,noc,mfa | 4 | 27 | 24 | 0 | 3 |
| dit,noc,noai,nomi | 4 | 7 | 1 | 0 | 6 |
| fanIn,fanOut,noai,nomi | 4 | 6 | 0 | 0 | 6 |
| fanIn,fanOut,noc,noai | 4 | 6 | 0 | 0 | 6 |
| fanIn,fanOut,noc,nomi | 4 | 6 | 0 | 0 | 6 |
| fanIn,noc,noai,nomi | 4 | 6 | 0 | 0 | 6 |
| fanOut,noc,noai,nomi | 4 | 6 | 0 | 0 | 6 |
| dit,fanIn,fanOut,noai,nomi | 5 | 6 | 0 | 0 | 6 |
| dit,fanIn,fanOut,noc,noai | 5 | 6 | 0 | 0 | 6 |
| dit,fanIn,fanOut,noc,nomi | 5 | 6 | 0 | 0 | 6 |
| dit,fanIn,noc,noai,nomi | 5 | 6 | 0 | 0 | 6 |
| dit,fanOut,noc,noai,nomi | 5 | 6 | 0 | 0 | 6 |
| fanIn,fanOut,noc,noai,nomi | 5 | 6 | 0 | 0 | 6 |
| dit,fanIn,fanOut,noc,noai,nomi | 6 | 6 | 0 | 0 | 6 |
| Total | | 659 | 293 | 1 | 365 |

Finally, 659 sub-data sets were formed, which contains 67 different features sets (column 3 of Table 8). Afterwards, all 659 sub data sets have been passed through three phases; dropping same instances, dropping inconsistent instances, and filtration.

**(4) Cleaning.** The next step is cleaning where identical instances in the data sets are eliminated since these instances are worthless and, sometimes, confusing for the model. Afterwards, inconsistent instances are also removed, since these are inconsistency in data sets (*Henderson-Sellers, 1995*). In inconsistency, the occurrences of all the metrics contain same values and contain dissimilar class tags.

Our objective is only to identify the anomaly on the data sets. This problem may possibly be addressed in four different ways. First option is to drop both the instances, consequently, information will be lost. Second option is to drop instances of minor class, due to this, data set will become more skewed. Third option is to drop instances of major class that resultantly produces less skewness, and in fourth option keep both instances resultantly

reciprocities negates the effect of each other. In this study, third option is applied to keep the impact minimum on the data sets.

**(5) Filtration.** Small data sets have been dropped where the number of instances are ≤100 in the filtration phase. This filter is applied for the reason to employ ten-fold cross validation, not including replacement, that is typically the situation with validation of the model. Consequently, 293 data sets are eliminated while using this filter.

**(6) Skewness ≤ 9:1.** The objective is to identify the skewed data sets out of 659 sub-data sets and drop them. Skewness has not been addressed further in this study.

Skewness shows faulty or free of fault occurrences that must contain a percentage of data sets to ≤90 and ≥10. The skewness filter is applied in the case where only 100 instances in the data set, then a minimum one record from both groups exists in the case where no hierarchical 10-fold cross validation that is replaced. After applying this filter onto all 659 sub-data sets only one sub-data set found skewed, which is hence eliminated.

Though remaining data sets are imbalance yet it is addressed in two folds, model building algorithm selection and performance measure selection. In first case, we use SVM which is usually the choice of modeling for imbalance data set (*Xing, Guo & Lyu, 2005*; *Elish & Elish, 2008b*; *Singh, Kaur & Malhotra, 2009*; *Di Martino et al., 2011*; *Yu, 2012*; *Malhotra, Kaur & Singh, 2010*). In the case of performance evaluation we use cross entropy which is, again independent of imbalancencess in the data set.

Lastly, after applying cleaning, filtering, and skewness, 659 data sets were decreased to 365, which are shown in the last columns of Table 8.

# EXPERIMENT AND RESULTS

## Experiment setup

**Dataset:** 365 preprocessed data sets, as depicted in Table 8.

**Tools:** R Language version 3.4.3 (*Rajnish & Bhattacherjee, 2005*) in R Studio 1.1.383 (*Rajnish & Bhattacherjee, 2006b*).

**Data Splitting technique:** Ten-fold stratified cross-validation without replacement. The stratified splitting maintains the ratio of classes in all the folds. Moroever, we reported the average results attained in the 10 folds.

**Classifiers Algorithm:** SVM is generally appreciated by the SFP community (*Xing, Guo & Lyu, 2005*; *Elish & Elish, 2008b*; *Singh, Kaur & Malhotra, 2009*; *Di Martino et al., 2011*; *Yu, 2012*), for its applicability on real-world applications, non-linear data coverage, and well generalization in high dimensional space. SVM is utilized for model building. Stratified splitting without replacement is done for ten-fold cross validation. Finally, we reported the average results computed in the 10-folds splitting in the form Cross entropy loss, Accuracy, F-Measure and AUC for all data sets.

**SVM parameters**: Gaussian kernel: The kernal has following equation:

$$k(x,y) = exp(-\frac{\|x-y\|}{2\sigma^2}) \tag{6}$$

It is a general-purpose kernel. It does not require any specific pattern of data. Moreover, SVM has been built and validated Gaussian kernel functions of the kernel and the best

model. Complete working of SVM model building and result collection is shown in Algorithm 1.

---

**Algorithm 1:** SVM Model Building and Cross Validation

| | |
|---|---|
| 1 | function SVMbuild (365 data sets); |
| | **Input** : 365 data sets |
| | **Output:** Cross entropy, Accuracy, F-Measure and AUC of the model in every data set |
| 2 | **for** *cur − Dset ←* 1 *to* 365 **do** |
| 3 |     curr-Dset← zScoreScale(cur-Dset) |
| 4 |     stratifiedSplit(cur-Dset, j←10) |
| 5 |     **for** *j ←* 1 *to* 10 **do** |
| 6 |         train-set ← cur-Dset[j-1] |
| 7 |         test-set ← cur-Dset[j] |
| 8 |         four-Models ←∅ |
| 9 |         *kernel ← Gaussianradial* |
| 10 |         SVMModel ← trainSVM(train-set, kernel) |
| 11 |         PerformanceMeasures ← CrossEntropy, Acc, F-Measure and AUC (SVMModel) |
| 12 |     **end** |
| 13 |     all-Dataset-PerformanceMeasures ← {all-Dataset-PerformanceMeasures} ∪ {PerformanceMeasures} |
| 14 | **end** |
| 15 | return all-Dataset-PerformanceMeasures |

---

The focus of this article is the viability of inheritance metrics in software fault prediction. It is found that the conventional techniques nevertheless being quite old, are still being used by the SFP community. The obvious reason is the outperformance of the discussed algorithms on the data set used. We didn't use the advanced machine learning techniques, explicitly. The reason is that they are accumulated using the algorithms discussed in the articles, therefore they are implicitly discussed in the article. Apart from that, advanced machine learning techniques are found more effective in huge data sets having a large number of features. Unfortunately, such data sets are seriously lacking in the Software engineering domain.

The experiment was conducted while keeping the industrial objective in mind where model is deployed with single threshold. Although model is supposed to be checked at every threshold but needs to be deployed with the threshold that gives the best results. So performance measure that visualize the performance of the model on that very threshold is of our interest, exactly where error entropy sits well, whereas AUC computes the models performance at every threshold, which does not suit with the objective.

## RESULTS AND DISCUSSION

There are facts that inheritance metrics are significantly different from other metrics. Firstly, semantic distinction of inheritance metrics from other metrics need no comments and justification. Secondly, data set view point, Rashid et al. (*Aziz, Khan & Nadeem, 2019*) proved the empirical distinction between inheritance, and non-inheritance metrics in his study. Moreover, inheritance metrics itself quite distinct from each other. This has been approved by visualizing the tendency of correlation between inheritance metrics. To illustrate this we compute the Pearson r, and the Spearman p correlation coefficient for the pairs discovered in forty selected public data sets. Correlation between the features of each

data set is computed in unfiltered data set. Nevertheless, all combinations are positively correlated, where not a single pair is equal to $\geq 0.7$ or $\leq -0.7$.

The fundamental objective of this work is to assess the exclusive viability of inheritance metrics in SFP, whereas the secondary aim is to achieve the greatest outcomes from the algorithms of machine learning. Keeping these in view, filtration is applied to data sets and experiments are planned.

### Overall cross entropy loss

In this context, Table 9 displays the outcomes of the experiments where the name of the feature is shown in the first column, the number of features in the second column, the overall number of data sets in the third column. Averages of Cross Entropy, Accuracy, F-Measure, and AUC are in column four, five, six and seven. The minimum values of Cross Entropy, Auuracy, F-measure, and AUC are in column eight, nine, ten, and eleven respectively. Cross-entropy loss, Accuracy, F-Measure, and AUC are computed for all 365 data sets comprising of 67 unique combinations. These unique combinations are plotted in Fig. 2 where the number of features are increasing from bottom to top. The upper part depicts the least mean entropy loss, average in the middle, and maximum at the bottom. This graph proves our objective that adding inheritance metrics will reduce the entropy loss. The findings are also validated by Accuracy, F-Measure, and AUC mentioned in Table 9.

Figure 2 graphically compares the cross entropy loss calculated through SVM on 1, 2, 3, 4, 5, and 6 feature sets of inheritance metrics on 365 data sets comprising 67 unique features. The lower part of Fig. 2 is heavy since the features are less. The results are gradually decreasing while moving upward since feature sets are increasing from 2 to 6. This graph proves our objective that adding inheritance metrics will reduce the entropy loss. The findings are also validated by Accuracy, F-Measure, and AUC.

Table 9 shows that overall {dit,ic,noc,mfa} achieved least entropy rate of 0.000723, and {fanIn, fanOut, noc, noai, nomi} achieved average least entropy rate of 0.001707.

Figure 3 shows the absence of outliers in the results across all performance measures. It can therefore be safely stated that the averages of performance measures are not biased.

### Feature wise cross entropy loss

Regarding the exclusive assessment of inheritance metrics in the context of SFP, which is the main goal of this article, adding inheritance metrics will reduce the Cross Entropy Loss. In this regard feature wise Cross Entropy Loss, Accuracy, F-Measure, and AUC are extracted from Table 9. The first column of Table 10 shows the feature set, number of features from 1 to 6 in the second column, and least Cross Entropy Loss in the third column. In order to support the findings Accuracy, F-measure, and AUC in column four, five, and six respectively. The overall findings are:

1. The results shown in Table 10 contain two distant feature sets, feature number 1 to 4, and 5 to 6. The first set comprises of {mfa, ic, noc, dit} and second set comprises of {dit, fanIn, fanOut, noc, noai, nomi}.

Aziz et al. (2021), *PeerJ Comput. Sci.*, DOI 10.7717/peerj-cs.563

**Table 9  Evaluation parameters for all features sets.**

| Features | # F | # Sets | Average | | | | Minimum | | | |
|---|---|---|---|---|---|---|---|---|---|---|
| | | | Cross entropy | Accuracy | F-Measure | AUC | Cross entropy | Accuracy | F-Measure | AUC |
| mfa | 1 | 3 | 0.00731 | 0.25978 | 0.40285 | 0.61196 | 0.00789 | 0.11111 | 0.20000 | 0.30210 |
| nomi | 1 | 3 | 0.00851 | 0.19259 | 0.31892 | 0.56063 | 0.00242 | 0.11765 | 0.21053 | 0.44420 |
| noai | 1 | 2 | 0.00916 | 0.40741 | 0.57895 | 0.48219 | 0.00895 | 0.40741 | 0.57895 | 0.39453 |
| ic,mfa | 2 | 3 | 0.00315 | 0.29366 | 0.43792 | 0.65072 | 0.00125 | 0.10256 | 0.18605 | 0.49415 |
| noc,mfa | 2 | 3 | 0.00341 | 0.24679 | 0.38847 | 0.64984 | 0.00123 | 0.12500 | 0.22222 | 0.44189 |
| dit,mfa | 2 | 3 | 0.00342 | 0.24493 | 0.38496 | 0.63859 | 0.00323 | 0.15000 | 0.26087 | 0.59120 |
| fanOut,nomi | 2 | 6 | 0.00359 | 0.28723 | 0.42839 | 0.68552 | 0.00829 | 0.17241 | 0.29412 | 0.32919 |
| fanIn,nomi | 2 | 6 | 0.00423 | 0.26969 | 0.40781 | 0.62949 | 0.00281 | 0.13043 | 0.23077 | 0.51891 |
| fanOut,noai | 2 | 5 | 0.00470 | 0.29504 | 0.44757 | 0.77786 | 0.00732 | 0.14286 | 0.25000 | 0.55104 |
| fanIn,fanOut | 2 | 6 | 0.00561 | 0.37622 | 0.53416 | 0.69704 | 0.00213 | 0.12245 | 0.21818 | 0.50322 |
| noc,nomi | 2 | 7 | 0.00581 | 0.32125 | 0.44381 | 0.65221 | 0.00116 | 0.11628 | 0.20833 | 0.43113 |
| noai,nomi | 2 | 8 | 0.00600 | 0.33470 | 0.45916 | 0.58975 | 0.00267 | 0.17308 | 0.29508 | 0.63020 |
| dit,nomi | 2 | 6 | 0.00661 | 0.37477 | 0.48181 | 0.67646 | 0.00710 | 0.19231 | 0.32258 | 0.83330 |
| fanIn,noai | 2 | 6 | 0.00673 | 0.25858 | 0.40530 | 0.68944 | 0.00447 | 0.11321 | 0.20339 | 0.34161 |
| fanOut,noc | 2 | 4 | 0.00754 | 0.32039 | 0.47248 | 0.90753 | 0.00351 | 0.10000 | 0.18182 | 0.31938 |
| noc,noai | 2 | 5 | 0.00780 | 0.38349 | 0.52902 | 0.73609 | 0.00660 | 0.22857 | 0.37209 | 0.57792 |
| dit,fanOut | 2 | 4 | 0.00784 | 0.33593 | 0.49815 | 0.66261 | 0.00408 | 0.10345 | 0.18750 | 0.31052 |
| dit,noai | 2 | 5 | 0.00826 | 0.41093 | 0.55152 | 0.55306 | 0.00901 | 0.20000 | 0.33333 | 0.39503 |
| fanIn,noc | 2 | 4 | 0.00881 | 0.26334 | 0.41176 | 0.52607 | 0.00479 | 0.13043 | 0.23077 | 0.56760 |
| dit,fanIn | 2 | 4 | 0.00919 | 0.28489 | 0.43365 | 0.65507 | 0.00504 | 0.17391 | 0.29630 | 0.34834 |
| dit,noc | 2 | 2 | 0.01093 | 0.47727 | 0.58974 | 0.65905 | 0.01030 | 0.18182 | 0.30769 | 0.57995 |
| fanIn,fanOut,nomi | 3 | 6 | 0.00206 | 0.26114 | 0.39801 | 0.72921 | 0.00448 | 0.13462 | 0.23729 | 0.39846 |
| fanOut,noai,nomi | 3 | 6 | 0.00263 | 0.26605 | 0.40493 | 0.71459 | 0.00252 | 0.15385 | 0.26667 | 0.52981 |
| fanOut,noc,nomi | 3 | 6 | 0.00278 | 0.27245 | 0.41109 | 0.79619 | 0.00249 | 0.14286 | 0.25000 | 0.40545 |
| fanIn,fanOut,noai | 3 | 6 | 0.00284 | 0.29089 | 0.43646 | 0.67128 | 0.00218 | 0.13333 | 0.23529 | 0.42610 |
| fanIn,noai,nomi | 3 | 6 | 0.00284 | 0.26575 | 0.40145 | 0.62098 | 0.00211 | 0.17500 | 0.29787 | 0.48502 |
| ic,mfa,noc | 3 | 3 | 0.00294 | 0.27869 | 0.42297 | 0.72094 | 0.00349 | 0.17910 | 0.30380 | 0.42501 |
| dit,mfa,noc | 3 | 3 | 0.00302 | 0.27185 | 0.41629 | 0.65053 | 0.00234 | 0.13208 | 0.23333 | 0.30763 |
| dit,fanOut,nomi | 3 | 6 | 0.00312 | 0.27066 | 0.40932 | 0.64407 | 0.00137 | 0.11111 | 0.20000 | 0.41411 |
| fanIn,noc,nomi | 3 | 6 | 0.00339 | 0.26596 | 0.40151 | 0.62571 | 0.00206 | 0.12727 | 0.22581 | 0.47119 |
| dit,fanIn,nomi | 3 | 6 | 0.00349 | 0.25965 | 0.39580 | 0.59709 | 0.00233 | 0.12500 | 0.22222 | 0.37093 |

**Table 9** (*continued*)

| Features | # F | # Sets | Average | | | | Minimum | | | |
|---|---|---|---|---|---|---|---|---|---|---|
| | | | Cross entropy | Accuracy | F-Measure | AUC | Cross entropy | Accuracy | F-Measure | AUC |
| dit,fanIn,fanOut | 3 | 6 | 0.00358 | 0.31639 | 0.46312 | 0.65336 | 0.00177 | 0.11864 | 0.21212 | 0.52079 |
| dit,fanOut,noai | 3 | 6 | 0.00405 | 0.25577 | 0.39723 | 0.62693 | 0.00088 | 0.11111 | 0.20000 | 0.37349 |
| noc,noai,nomi | 3 | 8 | 0.00405 | 0.30773 | 0.44233 | 0.64096 | 0.00182 | 0.12500 | 0.22222 | 0.42110 |
| fanIn,fanOut,noc | 3 | 6 | 0.00410 | 0.33394 | 0.48534 | 0.70194 | 0.00089 | 0.11364 | 0.20408 | 0.44540 |
| fanOut,noc,noai | 3 | 6 | 0.00417 | 0.25603 | 0.40023 | 0.69987 | 0.00173 | 0.12698 | 0.22535 | 0.60076 |
| dit,noai,nomi | 3 | 8 | 0.00435 | 0.29943 | 0.42878 | 0.72201 | 0.00087 | 0.12500 | 0.22222 | 0.41651 |
| fanIn,noc,noai | 3 | 6 | 0.00489 | 0.24448 | 0.38631 | 0.63796 | 0.00178 | 0.14815 | 0.25806 | 0.35226 |
| dit,noc,nomi | 3 | 8 | 0.00495 | 0.29684 | 0.42560 | 0.66426 | 0.00264 | 0.12245 | 0.21818 | 0.38422 |
| dit,fanOut,noc | 3 | 5 | 0.00503 | 0.25842 | 0.40627 | 0.65085 | 0.00241 | 0.08333 | 0.15385 | 0.31551 |
| dit,fanIn,noai | 3 | 6 | 0.00513 | 0.23834 | 0.37851 | 0.66118 | 0.00268 | 0.12245 | 0.21818 | 0.30099 |
| dit,fanIn,noc | 3 | 5 | 0.00590 | 0.23308 | 0.37191 | 0.58804 | 0.00765 | 0.14815 | 0.25806 | 0.35478 |
| dit,ic,mfa | 3 | 4 | 0.00613 | 0.26794 | 0.40688 | 0.62737 | 0.00727 | 0.17391 | 0.29630 | 0.53010 |
| dit,noc,noai | 3 | 7 | 0.00621 | 0.36134 | 0.50250 | 0.63620 | 0.00340 | 0.13953 | 0.24490 | 0.32523 |
| fanIn,fanOut,noai,nomi | 4 | 6 | 0.00177 | 0.25042 | 0.38530 | 0.69209 | 0.00218 | 0.15476 | 0.26804 | 0.44852 |
| fanIn,fanOut,noc,nomi | 4 | 6 | 0.00193 | 0.25367 | 0.38925 | 0.67488 | 0.00202 | 0.11290 | 0.20290 | 0.35545 |
| dit,fanIn,fanOut,nomi | 4 | 6 | 0.00196 | 0.25429 | 0.36958 | 0.73321 | 0.00129 | 0.10638 | 0.19231 | 0.34208 |
| fanOut,noc,noai,nomi | 4 | 6 | 0.00223 | 0.26055 | 0.39747 | 0.64410 | 0.00184 | 0.12308 | 0.21918 | 0.36006 |
| dit,fanIn,fanOut,noai | 4 | 6 | 0.00245 | 0.28168 | 0.42259 | 0.72720 | 0.00124 | 0.11458 | 0.20561 | 0.43630 |
| fanIn,fanOut,noc,noai | 4 | 6 | 0.00247 | 0.28242 | 0.42448 | 0.58455 | 0.00127 | 0.11828 | 0.21154 | 0.50334 |
| dit,fanOut,noai,nomi | 4 | 6 | 0.00248 | 0.26200 | 0.39904 | 0.74634 | 0.00199 | 0.12069 | 0.21538 | 0.34897 |
| fanIn,noc,noai,nomi | 4 | 6 | 0.00254 | 0.25246 | 0.38617 | 0.70396 | 0.00153 | 0.12676 | 0.22500 | 0.35997 |
| dit,fanOut,noc,nomi | 4 | 6 | 0.00254 | 0.26517 | 0.40303 | 0.78515 | 0.00163 | 0.11475 | 0.20588 | 0.33064 |
| dit,fanIn,noai,nomi | 4 | 6 | 0.00270 | 0.26010 | 0.39531 | 0.73084 | 0.00159 | 0.13043 | 0.23077 | 0.47058 |
| dit,ic,noc,mfa | 4 | 3 | 0.00279 | 0.27505 | 0.39122 | 0.73354 | 0.00072 | 0.12000 | 0.04348 | 0.50000 |
| dit,fanIn,noc,nomi | 4 | 6 | 0.00285 | 0.24841 | 0.38130 | 0.62960 | 0.00159 | 0.12857 | 0.22785 | 0.46006 |
| dit,fanIn,fanOut,noc | 4 | 6 | 0.00291 | 0.29058 | 0.43401 | 0.66073 | 0.00171 | 0.14063 | 0.24658 | 0.39800 |
| dit,noc,noai,nomi | 4 | 8 | 0.00331 | 0.51159 | 0.66989 | 0.85684 | 0.00211 | 0.13846 | 0.24324 | 0.34073 |
| dit,fanOut,noc,noai | 4 | 6 | 0.00333 | 0.30576 | 0.44797 | 0.72262 | 0.00194 | 0.16379 | 0.28148 | 0.57580 |
| dit,fanIn,noc,noai | 4 | 6 | 0.00376 | 0.22851 | 0.36565 | 0.66170 | 0.00162 | 0.40196 | 0.57343 | 0.78049 |
| fanIn,fanOut,noc,noai,nomi | 5 | 6 | 0.00172 | 0.24315 | 0.37650 | 0.74827 | 0.00120 | 0.11000 | 0.19820 | 0.38091 |
| dit,fanIn,fanOut,noc,nomi | 5 | 6 | 0.00185 | 0.24803 | 0.38232 | 0.54270 | 0.00122 | 0.11111 | 0.20000 | 0.35658 |
| dit,fanIn,fanOut,noai,nomi | 5 | 6 | 0.00186 | 0.24988 | 0.38492 | 0.69476 | 0.00127 | 0.11579 | 0.20755 | 0.42572 |
| dit,fanIn,fanOut,noc,noai | 5 | 6 | 0.00212 | 0.27066 | 0.40893 | 0.70901 | 0.00182 | 0.11765 | 0.21053 | 0.37438 |
| dit,fanOut,noc,noai,nomi | 5 | 6 | 0.00220 | 0.25810 | 0.39409 | 0.71781 | 0.00146 | 0.12329 | 0.21951 | 0.57075 |
| dit,fanIn,noc,noai,nomi | 5 | 6 | 0.00242 | 0.25452 | 0.38759 | 0.59626 | 0.00148 | 0.12658 | 0.22472 | 0.58879 |
| dit,fanIn,fanOut,noc,noai,nomi | 6 | 6 | 0.00174 | 0.24222 | 0.37580 | 0.77904 | 0.00118 | 0.10891 | 0.19643 | 0.61710 |

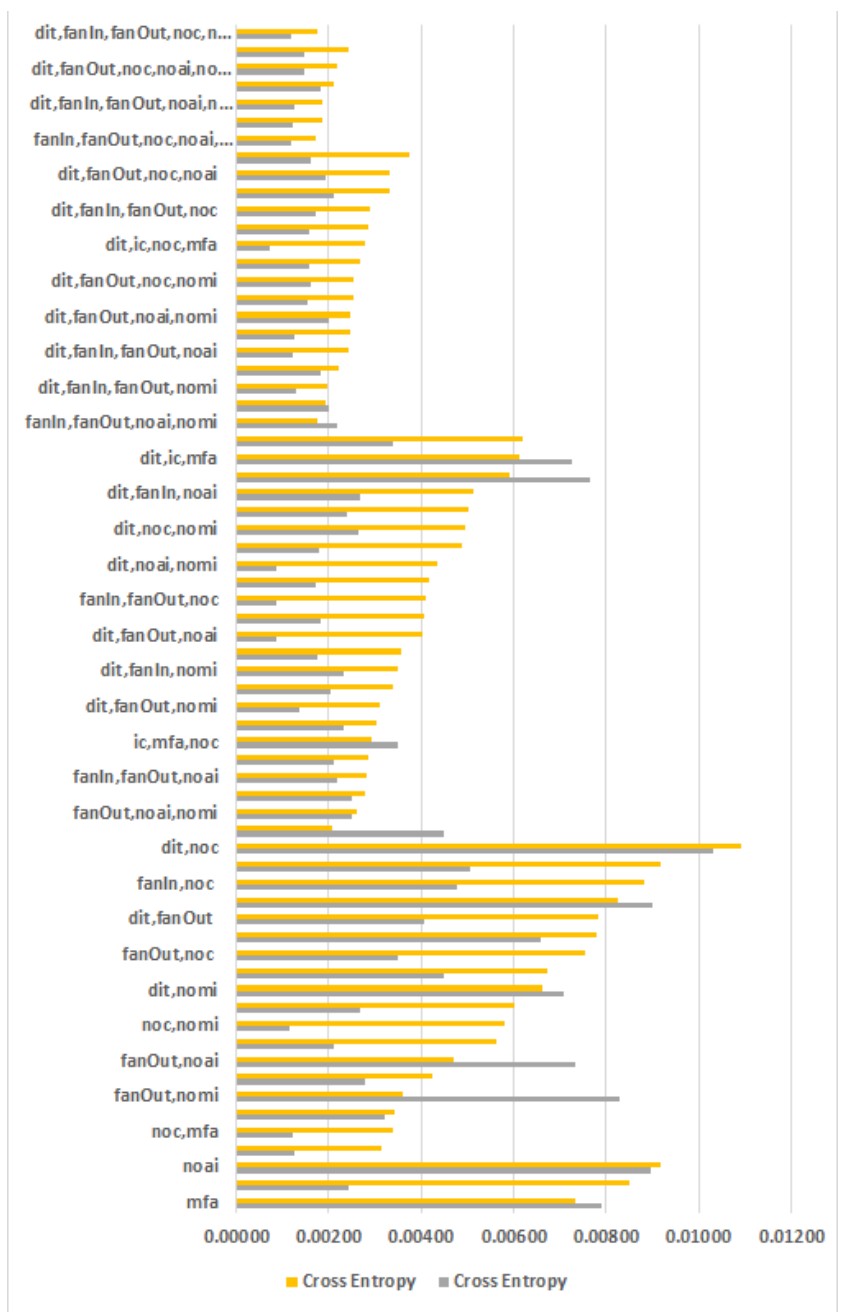

**Figure 2** Feature wise cross entry loss of all data sets.

2. Considering the single inheritance feature, {mfa} achieved the highest entropy rate of 0.0024225. After inserting another inheritance feature {ic} with {mfa} the error rate is further reduced to 0.0011558. Similarly adding {noc} with {ic, mfa} the entropy rate further reduced to 0.008679. The rate further reduced to 0.0007233 by adding {dit} into the existing combination of {mfa, ic, noc}.
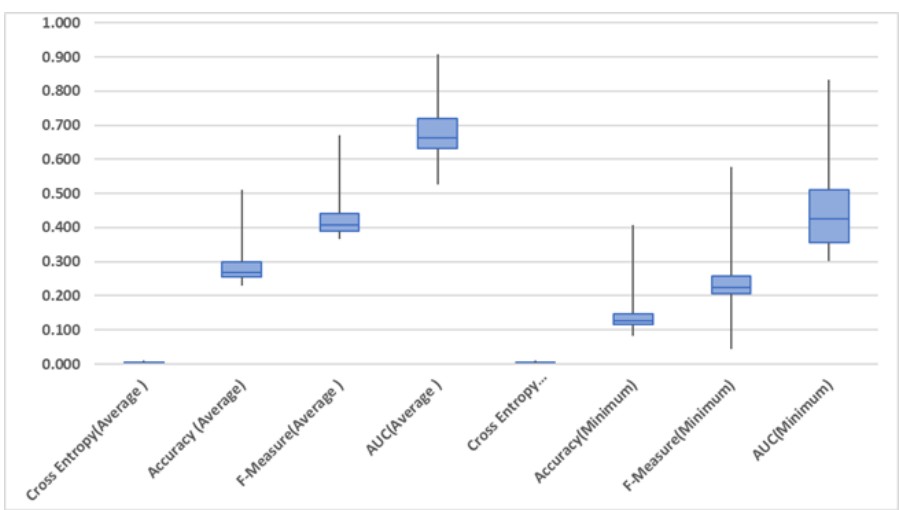

**Figure 3** Box plot of performance measures.

**Table 10  Feature wise minimum rate.**

| Feature set | # F | Cross Entropy | Accuracy | F-Measure | AUC |
|---|---|---|---|---|---|
| mfa | 1 | 0.0024225 | 0.1176500 | 0.2105300 | 0.4442000 |
| ic,mfa | 2 | 0.0011558 | 0.1162800 | 0.2083300 | 0.4311300 |
| ic,mfa,noc | 3 | 0.0008679 | 0.1250000 | 0.2222000 | 0.4165100 |
| dit,ic,noc,mfa | 4 | 0.0007233 | 0.1200000 | 0.0434800 | 0.5000000 |
| fanIn,fanOut,noc,noai,nomi | 5 | 0.0011961 | 0.1100000 | 0.1982000 | 0.3809100 |
| dit,fanIn,fanOut,noc,noai,nomi | 6 | 0.0011813 | 0.1089100 | 0.1964300 | 0.6171000 |

3. Considering feature set of five inheritance metrics {fanIn, fanOut, noc, noai, nomi} the Cross Entropy error rate is 0.0011961. When adding {dit} the r rate is dropped to 0.0011813.

The graphical representation of Table 10 is depicted in Fig. 4, which shows the results of two distant feature sets. The first set comprises of {mfa, ic, noc, dit}, and second set comprises of {dit, fanIn, fanOut, noc, noai, nomi}. Adding inheritance metrics into {mfa} the Cross Entropy rate is reduced significantly from 1 to 4 features sets. Similarly adding {dit} into the 5th set will further reduce the Entropy rate. The findings are also supported by the results of other performance measure Accuracy, F-measure, and AUC in Table 10.

*Feature wise average rate*

Regarding the overall exclusive assessment of inheritance metrics on to 365 data sets comprising 67 unique features, the average is calculated for all unique features to make an overall assessment. Table 11 shows the results where the first column contains a feature set, number of inheritance features in the 2nd column, and average score for Cross Entropy

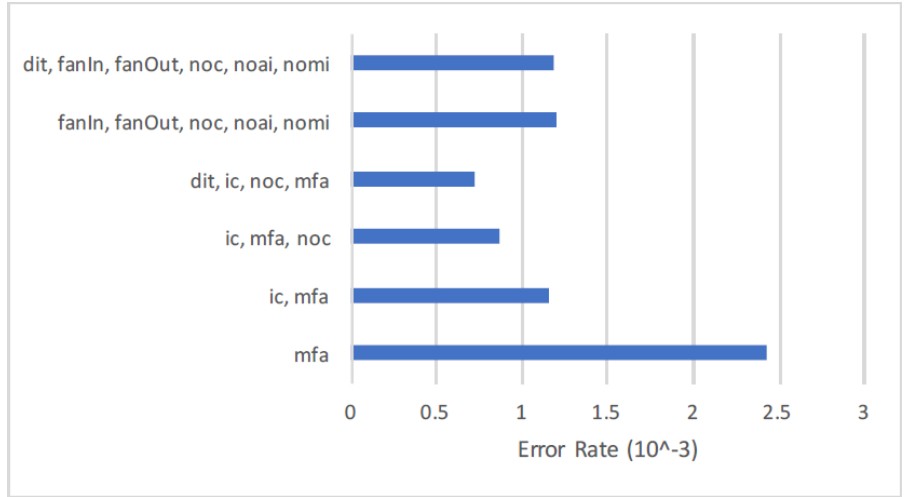

**Figure 4** Feature wise cross entropy rate.

**Table 11** Feature wise average rate.

| Feature set | # F | Cross Entropy | Accuracy | F-Measure | AUC |
|---|---|---|---|---|---|
| mfa | 1 | 0.007313 | 0.259780 | 0.402850 | 0.611960 |
| ic,mfa | 2 | 0.003154 | 0.293660 | 0.437920 | 0.650720 |
| fanIn,fanOut,nomi | 3 | 0.002065 | 0.261140 | 0.398010 | 0.729210 |
| fanIn,fanOut,noai,nomi | 4 | 0.001773 | 0.250420 | 0.385300 | 0.692090 |
| fanIn,fanOut,noc,noai,nomi | 5 | 0.001720 | 0.243150 | 0.376500 | 0.748270 |
| dit,fanIn,fanOut,noc,noai,nomi | 6 | 0.001743 | 0.242220 | 0.375800 | 0.779040 |

Loss in 3rd column. In order to support the findings Accuracy, F-measure, and AUC in column four, five and six respectively. The overall findings are:

1. The results are shown in Table 11 also contain two distant feature sets, feature number 1 to 2, and 3 to 6. The first set comprises on {mfa, ic} and second set comprises on {dit,fanIn,fanOut, noc, noai, nomi}.

2. Considering the single inheritance feature, {mfa} achieved the highest average of entropy rate of 0.0073134. After inserting another inheritance feature {ic} with {mfa} the average is reduced to 0.0031541.

3. Considering feature set of three inheritance metrics {fanIn, fanOut, nomi} the average entropy rate is 0.0020645. Adding {noai} the average entropy rate is reduced to 0.0017731. Similarly adding {noc} the average entropy rate is reduced to 0.0017595 and finally adding {dit} the rate further reduced to 0.0017428.

The graphical representation of Table 11 is depicted in Fig. 5, which shows the results of two distant feature sets. The first set comprises of {mfa,ic} and the second set comprises of {dit, fanIn, fanOut, noc, noai,{nomi}. Adding inheritance metrics into {mfa}, the entropy rate is reduced significantly from 1 to 2 features sets. Similarly adding {noai, noc} and {dit} into {fanIn, fanOut, nomi} will further reduce the error rate. The findings

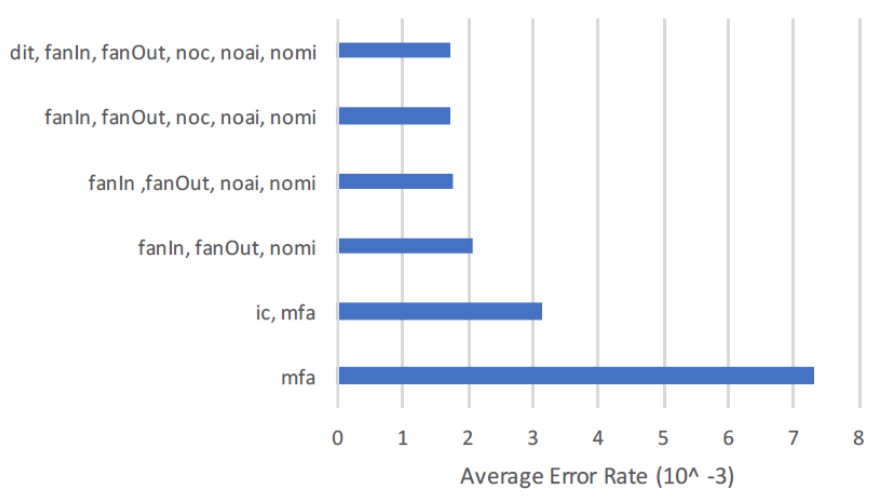

**Figure 5** Feature wise average.

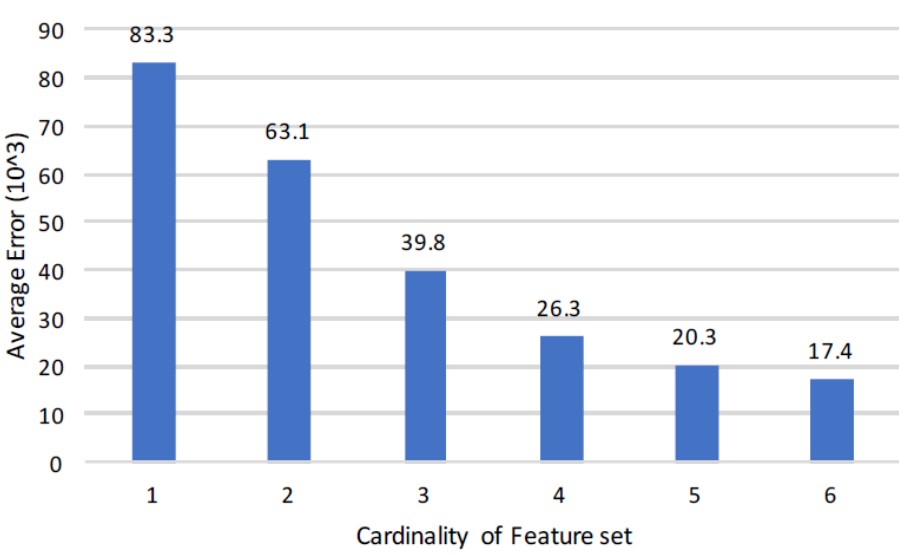

**Figure 6** Entropy error computed across the varying feature sets' cardinality.

are also validated by the results of other performance measure Accuracy, F-measure and AUC in Table 11.

### Cardinality of features

The cardinality of the feature set is 6 in this paper. Figure 6 shows the average Cross Entropy rate of a single feature set, double feature set, and up to 6 feature set. This clearly depicted, adding inheritance metrics will gradually reduce the entropy rate. The average entropy rate is 83.3 when single inheritance metrics is used. The average rate is reduced to 63.1 when two inheritance metrics are used. The average rate is further reduced to 39.8 when three

**Table 12 Cardinality of feature sets.**

| Cardinality of feature set | Average | | | | Minimum | | | |
|---|---|---|---|---|---|---|---|---|
| | Cross entropy | Accuracy | F-measure | AUC | Cross entropy | Accuracy | F-measure | AUC |
| 1 | 0.008327 | 0.286594 | 0.433572 | 0.551595 | 0.006422 | 0.212055 | 0.329825 | 0.380276 |
| 2 | 0.006312 | 0.321061 | 0.461427 | 0.668689 | 0.004722 | 0.147710 | 0.255616 | 0.486920 |
| 3 | 0.003984 | 0.277080 | 0.417081 | 0.664414 | 0.002659 | 0.133590 | 0.235035 | 0.416297 |
| 4 | 0.002626 | 0.280166 | 0.416391 | 0.705459 | 0.001642 | 0.144752 | 0.237041 | 0.438186 |
| 5 | 0.002029 | 0.254057 | 0.389057 | 0.668134 | 0.001409 | 0.117403 | 0.210084 | 0.449521 |
| 6 | 0.001743 | 0.242223 | 0.375799 | 0.779036 | 0.001181 | 0.108911 | 0.196429 | 0.617097 |

inheritance metrics are used. The average rate is gradually reduced further to 26.3, 20.3 and 17.4 when four, five and six inheritance metrics are used respectively.

Averages of minimum and average performance measures for single, double, triple and up to six feature set are calculated from Table 9 and depicts in Table 12. The first column shows the Cardinality of Feature set and column two, three, four and five contains averages of Cross entropy, Accuracy, F-Measure and AUC. The average of minimum values of Cross Entropy, Accuracy, F-Measue, AUC in column six, seven eight and nine. The graphical representation of Cross Entropy Loss is shown in Fig. 6 which proves our objects that adding inheritance matrices will reduce the cross entropy rate. Similarly, the results of Accuracy, F-Measure and AUC depicts in Table 12 also endorse these findings.

Furthermore, we conduct an experimental study to see how significantly metrics of inheritance assist in the prediction of software faults. Thus we made compassion of inheritance metrics and Chidamber and Kemerer (C&K) metrics suite. The findings demonstrate an appropriate impact of metrics of inheritance in software fault prediction (*Aziz, Khan & Nadeem, 2019*).

# THREATS, CONCLUSION AND FUTURE WORK

## Threats to validity

Our study relies on the data sets obtained from the repositories of tera-PROMISE, NASA and D'Ambros. In these repositories, insufficient information is available regarding the faults nor it indicates any certain type of software fault.

Primarily, faults are not indicated for the specific category of software fault. Therefore the projection might not be generalized for all categories of faults associated with the software. Likewise selected data set to encompass limited software products by diversification in team, design, scope, etc. The circumstance of fault might not be due to the inheritance aspect only.

Since the information associated with the projects are not available in the selected data sets, therefore, the identification of most associated factor to faults cannot be determined. Though the experiments do advocate the predictive ability of inheritance metrics yet the causation relationship may not be guaranteed.

We do not claim the general representation of the results across the algorithm. Yet the applicability of the result may conclude the mapping of independent variable over the dependent variable.

We selected SVM as a modeling algorithm, for being its general acceptance by the SFP community. Since the training process is heavily dependent on the data set available thus results may vary by varying the modeling algorithm. Likewise, results may also differ by varying the kernel function of SVM.

Lastly, selected metrics of inheritance are not covering every aspect related to inheritance in software products. Therefore generalization of selected metrics of inheritance might not possibly be the effect of every aspect of inheritance.

## Conclusion and future work

In this paper, we assessed software metrics of inheritance exclusively for their viability on software fault prediction. Experiments on forty distinct data sets argue inheritance metrics viability to faults prediction. The consensus of SVM revealed that inheritance metrics may accomplish the least entropy rate with a set of common characteristics. Overall {fanIn, fanOut, noc, noai, nomi} and individually {dit, ic, noc, mfa} proved to be the best predictor with the least entropy rate, whereas {dit}, {noc}, {ic} are helpful in reduction of entropy rate. We report that adding inheritance metrics is useful for predicting faults. These findings are also validated through performance measures of Accuracy, F-Measure, and AUC.

Regarding future work, we anticipate that some scholars may rebuild our experimentation and attempt to assess other metrics of inheritance than those we have employed. Since only nine metrics of inheritance are reviewed and assessed in this article and several other metrics of inheritance are defined in the literature. Due to the lack of availability of public data sets for these remaining metrics, these may not be assessed. This stimulates the requirement to build a data set that transmits data for the rest of the metrics. Also in this paper classification has been used due to less availability of continuous labels. So, the data set for the continuous value might be used for regression, to further evaluate the performance of inheritance metrics.

### Funding
The authors received no funding for this work.

### Competing Interests
The authors declare there are no competing interests.

### Author Contributions
- Syed Rashid Aziz, Tamim Ahmed Khan and Aamer Nadeem conceived and designed the experiments, performed the experiments, analyzed the data, performed the computation work, prepared figures and/or tables, authored or reviewed drafts of the paper, and approved the final draft.

## Data Availability

The raw data and the final calculation are available in the Supplementary Files.

## Supplemental Information

Supplemental information for this article can be found online at http://dx.doi.org/10.7717/peerj-cs.563#supplemental-information.

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
