# Peer review of "Exclusive use and evaluation of inheritance metrics viability in software fault prediction—an experimental study"

_PeerJ Computer Science, doi:10.7717/peerj-cs.563_

## Round 0.1 · original submission · Major Revisions

Although the topic covered in this manuscript and the proposed work are considered interesting, the reviewers have highlighted some critical issues that need to be overcome.

In particular, the first reviewer requests adding more details about the experimental part, on the use of SVM, and suggests considering some further state-of-the-art ML algorithms for SFP modeling.

The second reviewer requires a detailed literature survey on defect prediction. In addition, the authors should better justify their choices of considering inheritance metrics, rather than other metrics like cohesion or coupling, and of using SVM, rather than advanced ML techniques.

Moreover, a comparison between models developed using inheritance metrics and models developed using other metrics is required, in order to show the validity of the proposed work. Standard metrics such as accuracy, F-measure, and AUC value should be considered to show the effectiveness of the proposed models.

Reviewer 1 ·

Basic reporting

No Comment.

Experimental design

The experimental part needs improvement. Lots of effort have been devoted to the pre-processing step. However, details about model building using SVM are limited. Such as which kernel has been used. How the control parameters of SVM have been optimized. How dataset split into training and testing parts. Which error measure has been used. Why this particular measure was selected.

Further, using only one ML algorithm (SVM) is not enough. Authors should consider some state-of-the-art ML algorithms for SFP modeling.

Validity of the findings

In the context of Inheritence metrics, findings are sufficient, but in my view, comparison with the other OO metrics is limited and authors need to improve the paper in this part.

Reviewer 2 ·

Basic reporting

In this work, Authors have used and viability of inheritance metrics in SFP is evaluated through experiment. They have conducted a survey of inheritance metrics whose data is publicly available. They have collected about 40 data sets having inheritance metrics. Based on two criteria: these inheritance metrics have been cleaned and filtered; resulting in nine inheritance metrics captured. After preprocessing, the selected data sets have been divided into all possible combinations of inheritance metrics followed by merging similar metrics.

-The paper topic is good but the readability of this paper is very less.

Experimental design

1. Authors have not done detailed literatures survey on defect prediction. Authors should put the details of available techniques proposed by different researchers in the field on software defect predictions.
2. Presently, researchers in the filed of software engineering are using advance machine learning techniques to develop software fault prediction.
3. The authors have not given any idea about the reason of using only inheritance metrics i.e., why not cohesion or coupling

Validity of the findings

1. Authors are working on classification problem not on regression problem. Authors should compare the models using accuracy, F-measure, and AUC value
2. The authors should also prove that the developed models using inheritance metrics are significantly same of different from models developed using cohesion or coupling metrics.
3. Paper need detail threat to validity section.
4. Figure 2 is not easy to read. I recommend to use box-plot diagram.

Additional comments

In this work, Authors have used and viability of inheritance metrics in SFP is evaluated through experiment. They have conducted a survey of inheritance metrics whose data is publicly available. They have collected about 40 data sets having inheritance metrics. Based on two criteria: these inheritance metrics have been cleaned and filtered; resulting in nine inheritance metrics captured. After preprocessing, the selected data sets have been divided into all possible combinations of inheritance metrics followed by merging similar metrics.

-The paper topic is good but the readability of this paper is very less.



1. Authors have not done detailed literatures survey on defect prediction. Authors should put the details of available techniques proposed by different researchers in the field on software defect predictions.
2. Presently, researchers in the filed of software engineering are using advance machine learning techniques to develop software fault prediction.
3. The authors have not given any idea about the reason of using only inheritance metrics i.e., why not cohesion or coupling
4. Authors are working on classification problem not on regression problem. Authors should compare the models using accuracy, F-measure, and AUC value
5. The authors should also prove that the developed models using inheritance metrics are significantly same of different from models developed using cohesion or coupling metrics.
6. Paper need detail threat to validity section.
7. Figure 2 is not easy to read. I recommend to use box-plot diagram.


The paper is interesting, but needs *major* revision. The paper may be accepted, if the authors pay more attention to readability, clarity, and organization.

---

## Round 0.2 · Minor Revisions

The authors have addressed the reviewers' comments and the quality of the manuscript has improved.

I recommend the authors take into account the final suggestions of the first reviewer about Figures and Tables and revise the English of the manuscript with the support of an English expert.

Reviewer 1 ·

Basic reporting

A thorough proof-read of the paper is required. I advise the authors to use a language expert or language correction tool for the same.

Experimental design

No comment.

Validity of the findings

No comment.

Additional comments

The authors have addressed all my comments. There are a few comments about the figures and tables.
1. Many tables such as Table 1, Table 2, Table 5 are not clear. It seems that the authors have used images of the table. Please change them and use clear tables.
2. Similarly, Figures 2 and 3 are blurred. Change them.

A thorough proof-read of the paper is required. For example, in Abstract, sentences such as "Software Fault Prediction (SFP) is a way to spot faulty classes. It is helpful in the development and testing process. Software metrics are utilized in SFP." seem very immature. I advise the authors to use a language expert or language correction tool for the same.

---

## Round 0.3 · accepted · Accept

The authors have made the required minor revisions and the manuscript is now acceptable for publication. Congratulations to the authors!